# In situ single particle classification reveals distinct 60S maturation intermediates in cells

**Bronwyn A Lucas[1,2]\*, Kexin Zhang[1,2], Sarah Loerch[2†], Nikolaus Grigorieff[1,2]\***

[1]RNA Therapeutics Institute, University of Massachusetts Chan Medical School, Worcester, United States; [2]Howard Hughes Medical Institute, Janelia Research Campus, Ashburn, United States

**Abstract** Previously, we showed that high-resolution template matching can localize ribosomes in two-dimensional electron cryo-microscopy (cryo-EM) images of untilted *Mycoplasma pneumoniae* cells with high precision (Lucas et al., 2021). Here, we show that comparing the signal-to-noise ratio (SNR) observed with 2DTM using different templates relative to the same cellular target can correct for local variation in noise and differentiate related complexes in focused ion beam (FIB)-milled cell sections. We use a maximum likelihood approach to define the probability of each particle belonging to each class, thereby establishing a statistic to describe the confidence of our classification. We apply this method in two contexts to locate and classify related intermediate states of 60S ribosome biogenesis in the *Saccharomyces cerevisiae* cell nucleus. In the first, we separate the nuclear pre-60S population from the cytoplasmic mature 60S population, using the subcellular localization to validate assignment. In the second, we show that relative 2DTM SNRs can be used to separate mixed populations of nuclear pre-60S that are not visually separable. 2DTM can distinguish related molecular populations without the need to generate 3D reconstructions from the data to be classified, permitting classification even when only a few target particles exist in a cell.

**\*For correspondence:**
bronwyn.lucas@umassmed.edu (BAL);
niko@grigorieff.org (NG)

**Present address:** [†]Department of Chemistry and Biochemistry, University of California, Santa Cruz, United States

## Editor's evaluation

This paper explores the use of 2D high-resolution template-matching (2DTM) to locate and discriminate highly similar macromolecules within cryo-EM images of focused ion beam-milled cells. It demonstrates that differences in the 2DTM signal-to-noise ratios for located targets against multiple search templates can effectively segregate a mixed population of similar structures, as well as present a formal analysis strategy for probabilistic assignment of species within the mixed population. Because the identification of distinct structural states of macromolecular complexes inside the cell is a fundamental problem in 3D visual proteomics, this paper will be of broad interest to both structural and cell biologists.

## Introduction

Locating and characterizing molecules in cells is an important goal of molecular, structural, and cell biology. Cryogenic electron microscopy (cryo-EM) enables simultaneous visualization of all cellular molecules in their native cellular environment while preserving high-resolution molecular architecture. Therefore, cryo-EM holds the promise of delivering an atomistic view of the cell. However, realizing this promise is limited by the high density of molecules in a cell, and the low signal-to-noise ratio of cryo-EM micrographs, making it difficult to identify molecules of interest (*Lučič et al., 2013*). To address this, electron cryo-tomography (cryo-ET) can be used to build 3D maps of cellular structures in

their native context (in situ) by reconstructing tomograms from a series of images of tilted 2D samples (*Kürner et al., 2005*; *Lučič et al., 2013*; *Mahamid et al., 2016*). In a tomogram, molecules overlapping in any given view can be separated and large molecular assemblies (particles) with distinctive shapes can be identified. Once identified, subtomogram averaging can yield in situ molecular structures at <4 Å resolution (*Himes and Zhang, 2018*; *Tegunov et al., 2021*). However, since the effective resolution of a raw tomogram is below 15–20 Å (*Vilas et al., 2020*), identification of specific targets in tomograms is limited to abundant particles that are sufficiently distinct at this resolution to be identified.

Many potential cell biological applications require accurate categorization of individual molecule identity at a specific subcellular localization. Examples are the characterization of the spatial organization of a biosynthetic process such as ribosome biogenesis, and the assignment of molecular identities in small volumes such as synapses and vesicles. 3D classification of subtomograms can differentiate between structural states (*Himes and Zhang, 2018*; *Xue et al., 2021*). However, the assignment of states is unreliable for similar structures that can only be distinguished using high-resolution detail, and statistical approaches to quantitatively assess classification results are lacking. Machine learning has been employed for particle classification in tomograms, but currently only performs as well as a human operator (*Moebel et al., 2021*). While machine learning algorithms perform better than 3D template matching at molecule localization in tomograms, classification remains challenging for all algorithms (*Gubins et al., 2020*). In situ molecule classification, therefore, remains a major challenge.

We recently described an alternate method to locate particles that may improve structural classification in cells. By using 2D cryo-EM images, rather than tomograms, and fine-grained, high-resolution template matching (2DTM), specific particles can be located in cells with high precision using their atomic structures (*Lucas et al., 2021*; *Rickgauer et al., 2020*; *Rickgauer et al., 2017*). 2DTM uses molecular models, from in vitro structure determination or in silico structure prediction (e.g. Alphafold2 *Jumper et al., 2021*) to generate a 3D density. This 3D density (hereafter referred to as the template) is then used to calculate millions of 2D projections representing different orientations of the molecule. A pixel-wise cross-correlation of the 2D projections with a high-resolution 2D cryo-EM image is performed, yielding a 2DTM signal-to-noise ratio (SNR) of the best matching projection at every pixel location (*Rickgauer et al., 2017*). The 2DTM SNR values are subjected to a significance test by comparison to a Gaussian noise distribution following a matched filter, which is used to establish a threshold allowing a given number of false positives (*Lucas et al., 2021*; *Rickgauer et al., 2017*). In the following, we refer to targets passing this test as significant targets (*Lucas et al., 2021*; *Rickgauer et al., 2017*).

The 2DTM SNR is proportional to template mass and negatively affected by non-matching elements between template and target (*Lucas et al., 2021*; *Rickgauer et al., 2020*; *Rickgauer et al., 2017*). We have shown that a template generated from a *Bacillus subtilis* 50S large ribosomal subunit was able to detect 50S in 2D cryo-EM images of *Mycoplasma pneumoniae* cells, but with a lower average 2DTM SNR compared to a *M. pneumoniae* 50S template (*Lucas et al., 2021*). This demonstrated that (1) 2DTM using partially matching templates can be sufficiently sensitive to yield significant targets and (2) the mean 2DTM SNR of detected targets provides a read-out of the relative similarity between different templates and populations of particle species.

In this study, we investigate whether the ratio of 2DTM SNRs obtained using different templates can be used to identify the template that more closely resembles each cellular target, and thereby classify single particles in cells. As a model system, we chose to examine the late stages of 60S ribosomal subunit biogenesis in the yeast *Saccharomyces cerevisiae* because (1) intermediates are of a similar size and share significant structure with one another, making them difficult to separate at low resolution, (2) molecular models spanning multiple late intermediate states have recently been described, and (3) the maturation events that occur before and after nuclear export have been characterized. Subcellular localization can thereby validate the assignment of intermediate and mature states.

We show that 2DTM can locate and distinguish nuclear intermediates of 60S maturation in 2D cryo-EM images of focused ion beam (FIB)-milled yeast cells. We confirm that 2DTM can distinguish predefined 60S populations separated by subcellular localization and identify compositional differences between them. We apply a maximum likelihood-based approach to identify two sub-populations of nuclear intermediates that were not otherwise separable and provide a confidence of single particle

classification. We show that using this approach, we can observe a shift in the nuclear pre-60S intermediate population to a more mature intermediate after inhibiting Crm1-mediated nuclear export. This study demonstrates that different particle populations in cells can be modelled with a Gaussian distribution of their relative 2DTM SNR ratios to effectively distinguish related complexes and identify population changes in cells.

## Results

### 2DTM identifies 60S in biologically relevant locations and orientations in FIB-milled lamellae

2DTM has been used to detect mammalian ribosomes in thin extensions of adherent cells (*Rickgauer et al., 2020*), and bacterial ribosomes in *Mycoplasma pneumoniae* cells (*Lucas et al., 2021*), both of which are sufficiently thin to permit imaging by transmission EM (TEM). Since most eukaryotic cells are too thick to image by TEM, FIB-milling is used to generate thin, electron-transparent lamellae of cryogenically frozen cells (*Marko et al., 2007*; *Rigort et al., 2012*; *Villa et al., 2013*).

To evaluate the utility of 2DTM to locate molecules in FIB-milled lamellae, we collected 28 2D cryo-EM images of the nuclear periphery of 7 lamellae generated from actively growing *Sacchromyces cerevisiae* cells (*Figure 1*, *Figure 1—figure supplement 1A-B*)(*Table 1*). We identified 4363 large ribosomal subunits by 2DTM using a template generated from a model representing the mature 60S (PDB: 6Q8Y) (*Tesina et al., 2019*; *Figure 1A–C*). The peaks corresponding to significant detections were clearly distinguishable from background (*Figure 1D and E*, *Figure 1—figure supplement 1C*), enabling precise localization of mature 60S in the cell.

To assess the specificity of 60S detection, we identified regions of the images corresponding to the cytoplasm, nucleus and vacuole by visual inspection. Consistent with the expected high specificity of 2DTM, we did not observe any significant mature 60S-detected targets in regions of the image corresponding to the vacuole (*Figure 1C and G*). In contrast, 229 mature 60S-detected targets localized to the nucleus, representing ~5% of all mature 60S identified targets in these images, well above the expected one false positive per image (*Figure 1C–G*). We found that the normalized maximum intensity projections (MIPs) have a distribution of values similar in shape and location of their maxima in different regions of the image, corresponding to the nucleus, cytoplasm, and vacuole. This indicates that the probability of a false positives is also similar in these different compartments (*Figure 1—figure supplement 2A-E*). The slight differences in the MIP distributions between the compartments likely result from differences in their composition. We conclude that false detections are not more likely in visibly darker or denser regions of the image.

In regions of the images corresponding to the cytoplasm we observe a median density of ~6500 60S/$\mu m^3$, which, assuming an average cell volume of ~42 $\mu m^3$ of which ~65% is cytoplasm, corresponds to a total of ~180,000 60S/cell (*Figure 1H*). This is consistent with prior estimates of 187,000 ± 56,000 ribosomes per yeast cell based on rRNA concentration (*von der Haar, 2008*). The median density in the nucleus was ~30-fold lower at ~200 60S/$\mu m^3$ (*Figure 1H*). This agrees with the previously observed lower density of 60S particles in the nucleus, e.g. (*Delavoie et al., 2019*), but likely only reflects a subset of the nuclear 60S population.

Beyond the subcellular distribution of mature 60S-detected targets, we also confirmed that 2DTM identified specific 60S in biologically relevant locations and orientations. The nuclear envelope (NE) is contiguous with the endoplasmic reticulum and a known site for co-translational transport of transmembrane and secretory proteins, while the vacuole is not known to be a site of translation. We found that mature 60S-detected targets were oriented with their polypeptide exit tunnels facing the cytoplasmic surface of the NE but were depleted from within ~20 nm of the vacuole (*Figure 1C and F*). This indicates that the orientation of 60S identified by 2DTM is unlikely to be an artefact introduced by features of the membrane in the image.

To confirm that the targets identified with the mature 60S template reflect ribosomes, we generated a 3D reconstruction using the locations and orientations of 3991 significant mature 60S-detected targets using standard single particle approaches as described previously (*Lucas et al., 2021*). In addition to the 60S, the 10 Å-filtered reconstruction shows density reminiscent of the 40S small ribosomal subunit (*Figure 1I*). This is consistent with many of the mature 60S detected targets representing a population of 80S ribosomes. Local resolution estimation shows that the resolution of the 40S is lower

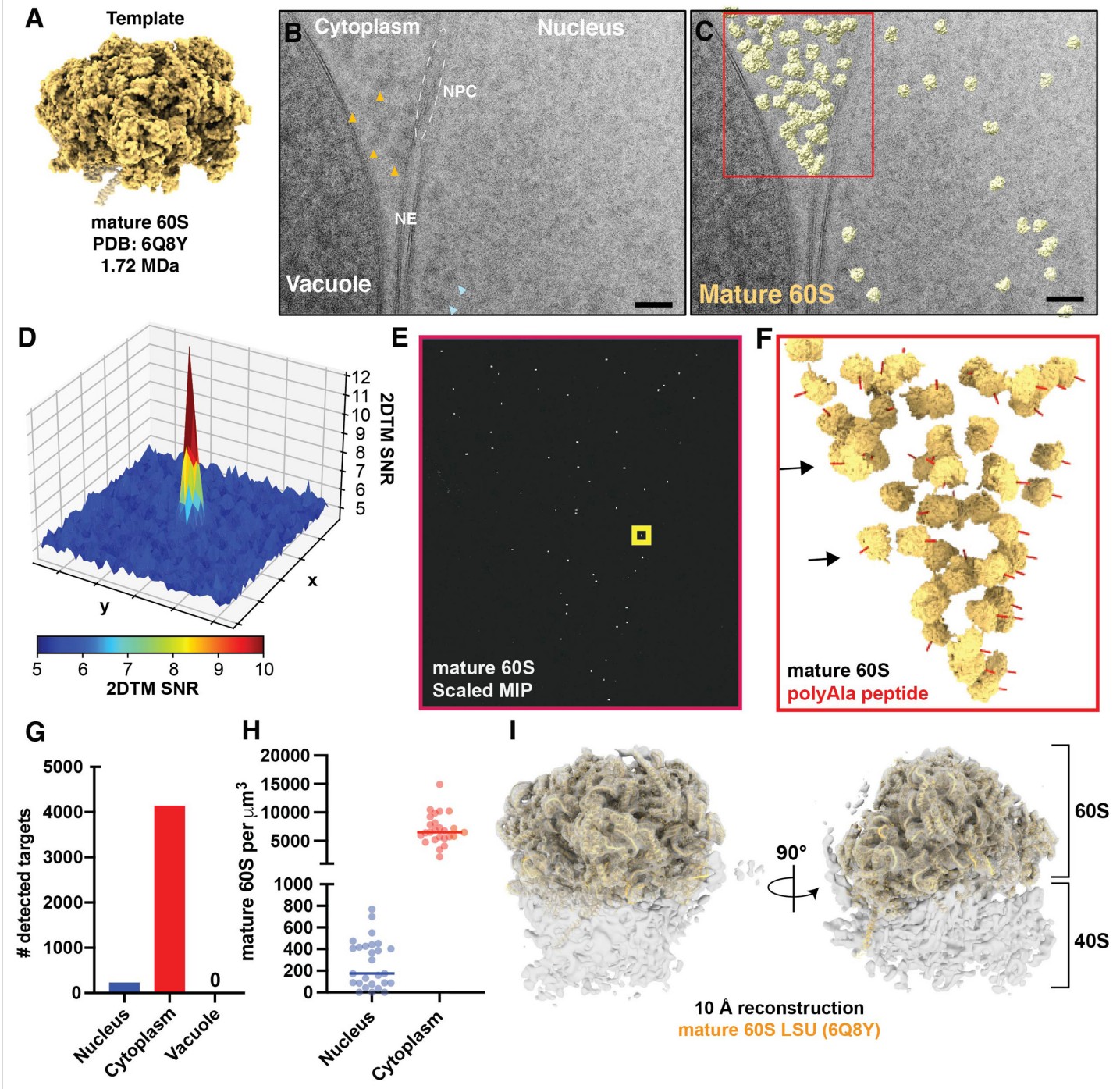

**Figure 1.** Detection of cytoplasmic mature 60S and mitochondrial ribosomes in 2D images of FIB-milled yeast lamella. (**A**) Cryo-EM like density generated using the atomic coordinates of PDB: 6Q8Y that correspond with the mature 60S. (**B**) TEM image of the nuclear periphery from a FIB-milled yeast lamella. Yellow arrows indicate low-resolution features in the cytoplasm that may indicate the presence of ribosomes. Blue arrows indicate regions of similar size and contrast in the nucleoplasm. NE: nuclear envelope; NPC: nuclear pore complex. (**C**) Cryo-EM micrograph of yeast nuclear periphery from FIB-milled lamella with the results from a 2DTM search using the mature 60S template. Significant targets are indicated by mapping the template in the best matching locations and orientations (shown in yellow). The red box indicates the regions highlighted in (E) and (F). Scale bar = 50 nm. (**D**) 3D surface representation showing the pixel-wise 2DTM SNRs in the 50x50 pixel region of the normalized maximum intensity projection (MIP) indicated by the yellow box in (E). Each square represents 10x10 pixels. Colors represent the SNR value of each pixel as indicated by the scale bar below. (**E**) Normalized MIP showing the results of 2DTM using the template in (A) in the region of (C) indicated in red. (**F**) 3D slab indicating the locations and orientations of mature 60S-detected targets in the indicated region of (C). The red polypeptide indicates the location of the polypeptide exit tunnel

*Figure 1 continued on next page*

*Figure 1 continued*

on each 60S. (**G**) Bar chart indicating the number of mature 60S-detected targets identified in the indicated subcellular compartments in 28 images of the nuclear periphery. (**H**) Plot showing the density of mature 60S in the regions of the images corresponding to the nucleus (blue) or cytoplasm (red). Each dot represents a different image. The solid bar indicates the median. (**I**) 10 Å filtered 3D reconstruction calculated from 3991 60S subunits at the locations and orientations detected in 28 images, showing clear density for the 40S small subunit. The molecular model of the 60S used to generate the template in (A) is shown in yellow.

The online version of this article includes the following figure supplement(s) for figure 1:

**Figure supplement 1.** FIB-milled yeast cells and detection statistics relative to a Gaussian noise model.

**Figure supplement 2.** Background 2DTM SNRs in the nucleus and vacuole relative to the cytoplasm.

**Figure supplement 3.** Reconstruction using 60S coordinates shows features consisitent with detection of 80S ribosomes.

**Table 1.** Estimated thickness and calculated defocus per image.

| Lamella # | Image file | Est thickness (nm) | Defocus 1 (Å) | Defocus 2 (Å) | Angle |
|---|---|---|---|---|---|
| 1 | 24_Mar11_13.30.44_1_0.mrc | 179 | 5577 | 5434 | 7.9 |
| 1 | 25_Mar11_13.32.53_3_0.mrc | 170 | 5943 | 5636 | 53.07 |
| 2 | 50_Mar11_15.04.14_36_0.mrc | 132 | 3626 | 3298 | 23.33 |
| 2 | 51_Mar11_15.07.03_38_0.mrc | 109 | 3944 | 3863 | 16.91 |
| 2 | 52_Mar11_15.10.54_40_0.mrc | 117 | 6491 | 6192 | 10.89 |
| 2 | 53_Mar11_15.21.33_42_0.mrc | 205 | 4906 | 4610 | 5.32 |
| 2 | 54_Mar11_15.24.09_44_0.mrc | 205 | 7701 | 6828 | –76.32 |
| 2 | 55_Mar11_15.26.04_46_0.mrc | 212 | 5141 | 4794 | 23.45 |
| 2 | 56_Mar11_15.33.15_48_0.mrc | 170 | 7240 | 7093 | 82.03 |
| 2 | 57_Mar11_15.36.27_50_0.mrc | 178 | 5969 | 5807 | –7.87 |
| 2 | 58_Mar11_15.38.03_52_0.mrc | 186 | 6445 | 6182 | –75.84 |
| 3 | 115_Mar12_10.39.05_93_0.mrc | 136 | 3436 | 3230 | –22.72 |
| 3 | 118_Mar12_10.46.25_99_0.mrc | 141 | 3291 | 3200 | 39.71 |
| 4 | 131_Mar12_11.32.55_127_0.mrc | 114 | 4532 | 4377 | 15.85 |
| 4 | 133_Mar12_11.37.56_131_0.mrc | 130 | 3252 | 2979 | 7.24 |
| 4 | 135_Mar12_11.42.53_135_0.mrc | 93 | 3041 | 2977 | –76.83 |
| 4 | 138_Mar12_11.50.12_141_0.mrc | 106 | 3497 | 3404 | 79.97 |
| 5 | 141_Mar12_11.57.31_147_0.mrc | 161 | 6153 | 5600 | –33.24 |
| 5 | 143_Mar12_12.02.15_151_0.mrc | 152 | 4552 | 4457 | –36.2 |
| 5 | 146_Mar12_12.19.02_157_0.mrc | 98 | 3902 | 3868 | 61.58 |
| 5 | 147_Mar12_12.21.27_159_0.mrc | 103 | 4305 | 4066 | 16.81 |
| 5 | 148_Mar12_12.23.52_161_0.mrc | 73 | 3728 | 3604 | –36.33 |
| 5 | 149_Mar12_12.26.15_163_0.mrc | 78 | 3696 | 3573 | 0.37 |
| 5 | 150_Mar12_12.28.45_165_0.mrc | 98 | 3723 | 3607 | 1.6 |
| 5 | 151_Mar12_12.31.16_167_0.mrc | 114 | 3771 | 3648 | 54.64 |
| 6 | 168_Mar12_13.11.12_199_0.mrc | 147 | 3157 | 3040 | –57.66 |
| 6 | 171_Mar12_13.18.43_205_0.mrc | 90 | 2309 | 2102 | 31.47 |
| 7 | 6 A_Mar11_14.59.49_34_0.mrc | 126 | 5210 | 4391 | 2.1 |

relative to the 60S (*Figure 1—figure supplement 3B*). This follows the expected positional heterogeneity of the 40S relative to the 60S when capturing ribosomes in a range of translation states (*Freitas et al., 2021*; *Korostelev, 2022*) and prior results using 50S *Mycoplasma pneumoniae* targets (*Lucas et al., 2021*). We conclude that 2DTM-identified locations and orientations in 2D cryo-EM images of FIB-milled lamellae reflect biologically relevant locations and orientations of ribosomes in the cell.

## Relative 2DTM SNRs enable single particle classification in situ

The nuclear envelope (NE) creates a physical barrier that separates premature 60S in the nucleus from mature 60S in the cytoplasm and is easily distinguishable in many 2D images by its characteristic double membrane and by the more granular appearance of the cytoplasm vs the nucleus (e.g. *Figure 1B*). Our observation of a substantial population of mature 60S-detected targets in the nucleus, but not in the vacuole (*Figure 1C and G*), suggests that the nuclear 60S may result from cross-detection of nuclear precursors, which share part of their structure with mature 60S and therefore also produce significant correlations (*Figure 2A*). As a first step to differentiate between related 60S intermediates, we located precursor 60S by 2DTM searches using a template generated from a late nuclear intermediate (LN 60S, PDB: 6N8J) (*Zhou et al., 2019*; *Figure 2A and B*), and annotated each target by its subcellular localization. The LN 60S was chosen because it represents the most mature nuclear intermediate for which there is a structure, and which retains ribosome biogenesis factors (RBFs) that are removed during nuclear and early cytoplasmic processing (*Figure 2A*). Thus, we expect that (1) the similarities between the mature 60S and LN 60S structures will result in cross-detection of the respective other complex and (2) the cytoplasmic population will more closely resemble the mature 60S and nuclear population will more closely resemble the LN 60S resulting in a higher mature 60S / LN 60S 2DTM SNR ratio in the cytoplasm than the nucleus. In the 28 images of the nucleus and nuclear periphery we located 1651 significant LN 60S-detected targets of which 1382 (~84%) of the LN 60S-detected targets were cytoplasmic and 268 (16%) were nuclear, a three-fold greater proportion than the mature 60S-detected targets (*Figure 2—figure supplement 1A*). We identified more cytoplasmic than nuclear targets in 2DTM searches with both mature and LN 60S templates because (1) the cytoplasm represented a larger area of our images and (2) the concentration of 60S is expected to be higher in the cytoplasm relative to the nucleus. Only one of the significant LN 60S-detected targets localized to the vacuole, which is below the expected false positive rate and further indicates the specificity of 2DTM.

As expected from the similarity between the mature and LN 60S templates, the locations of many of the targets identified in the two searches overlap (*Figure 2B and C*). We aligned the two sets of coordinates using the program *align_coordinates* (*Lucas et al., 2021*). Approximately one third of the mature 60S-detected targets overlapped with LN 60S-detected targets while 92% of the LN 60S-detected targets overlapped with mature 60S-detected targets (*Figure 2H*). Combining the results of both searches, 30% of the nuclear targets were LN 60S-detected only, compared to only 0.5% of the cytoplasmic targets (*Figure 2I*, *Figure 2—figure supplement 1A*).

Consistent with their expected localizations, the median $\log_2$(mature 60S / LN 60S 2DTM SNR) values of targets identified with both templates were significantly higher for the cytoplasmic population than the nuclear population ($P<0.0001$, K-S. test) (*Figure 2D–G and J*). We classified each target as LN or mature 60S according to the highest 2DTM SNR (*Figure 2K*). Of the population detected with both mature and LN 60S templates, 94% of the 1361 cytoplasmic targets have a closer match (higher SNR) with the mature 60S and 88% of the 171 nuclear targets have a closer match with the LN 60S (*Figure 2J*). Combining all 60S-detected targets, the nuclear 60S targets are now more clearly distinguished from the cytoplasmic population with 98% of the cytoplasmic targets annotated as mature 60S, and 60% of the nuclear targets annotated as pre-60S (*Figure 2K and L*). The ~40% of nuclear targets that more closely resemble the mature 60S likely reflect nuclear intermediates different from the LN 60S (see below) and thus do not perfectly match either template. We conclude that comparing 2DTM SNRs can effectively differentiate populations of related particles in situ.

## Defining a confidence metric for single particle classification in situ

To gain an understanding of cell biology at molecular resolution it is necessary to be able to confidently assign particle identity to individual targets. We show above that the nuclear and cytoplasmic 60S populations were significantly different with respect to their relative similarity to the LN and

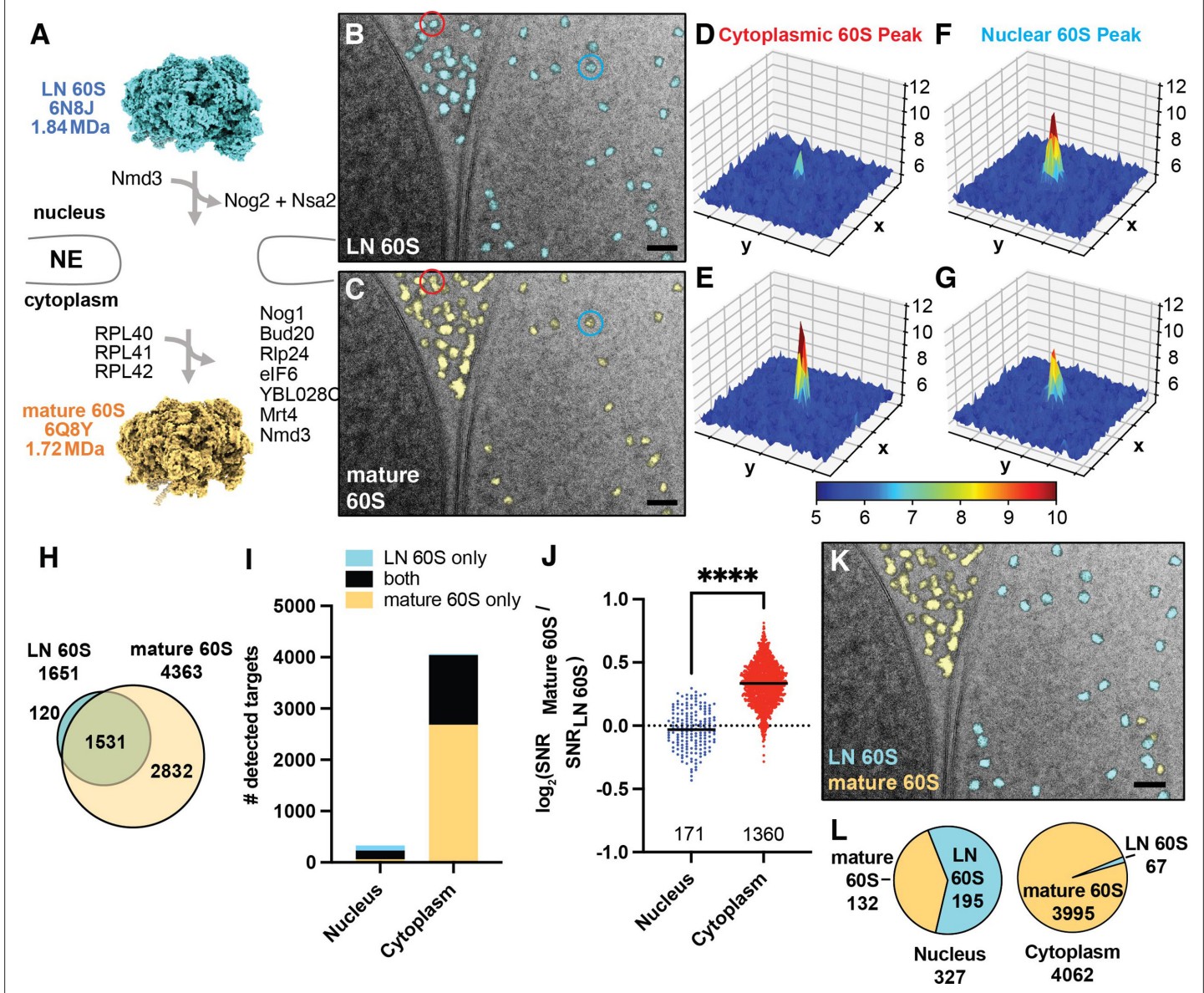

**Figure 2.** 2DTM SNRs differentiate cytoplasmic mature 60S from nuclear pre-60S in 2D images of FIB-milled yeast lamella. (**A**) Diagram showing the compositional changes that accompany the maturation from the late nuclear (LN) 60S (PDB: 6N8J), shown in blue, to the mature 60S (PDB: 6Q8Y), shown in yellow, in the cytoplasm. (**B**) Cryo-EM micrograph of yeast nuclear periphery from FIB-milled lamella with the results from a 2DTM search using the LN 60S template. Significant targets are indicated by mapping a projection of the template in the best matching locations and orientations (shown in blue). Scale bar = 50 nm. (**C**) As in (B), showing the results from a 2DTM search of the indicated image using the mature 60S as a template (yellow). (**D**) 3D surface representation showing the results of a 2DTM search with the LN 60S template in the 50x50 pixel region of the image in (B) highlighted in red. Each square represents 10x10 pixels. Colors represent the SNR value of each pixel as indicated by the scale bar below. (**E**) As in (D) showing the results of a 2DTM search with the mature 60S template in the corresponding region of the image in (C) highlighted in red. (**F**) As in (D) showing the results of a 2DTM search with the LN 60S template in the region of (B) highlighted in blue. (**G**) As in (D) showing the results of a 2DTM search with the mature 60S template in the corresponding region of the image in (C) highlighted in blue. (**H**) Diagram indicating the number of mature 60S (yellow) and LN 60S (blue)-detected targets identified in 2DTM searches of 28 images of the nuclear periphery. The overlap of the Venn diagram indicates the number of targets identified in both searches. (**I**) Bar chart indicating the number of targets detected by the mature 60S (yellow), the LN 60S (blue), and by both (black) in regions of the images corresponding to the nucleus or cytoplasm. (**J**) Plot showing the $\log_2$ 2DTM SNR ratios for LN and mature 60S-detected targets grouped by subcellular compartment. Each dot indicates a 60S detected in both searches. ****: $p < 0.0001$. (**K**) Image showing the identified targets color-coded by the best-matching template (blue: LN 60S, yellow: mature 60S) as determined by the higher 2DTM SNR at each overlapping location. Scale bar = 50 nm. (**L**) Pie chart indicating the proportion of all nuclear (left) and cytoplasmic (right) 60S targets that more closely resemble the mature 60S (yellow) or LN 60S (blue) template, respectively, as determined by the highest 2DTM SNR at each identified location and orientation.

The online version of this article includes the following figure supplement(s) for figure 2:

*Figure 2 continued on next page*

*Figure 2 continued*

**Figure supplement 1.** 2DTM SNR ratios distinguish the nuclear from the cytoplasmic 60S population.

mature 60S (*Figure 2*). We also show that classifying targets by their highest 2DTM SNR effectively separates the nuclear from the cytoplasmic population (*Figure 2*). However, a single threshold does not fully capture the differences between the nuclear and cytoplasmic populations and for an individual particle the confidence of classification is unclear.

To assign a confidence in the class assignments of detected particles we developed a likelihood-based approach to infer the probability of a particle deriving from one of a given number of populations. We sought to classify each of the 1531 LN and mature 60S-detected targets without prior knowledge of their subcellular localization. We restricted our analysis to the targets that were detected by both templates to limit the contribution from noise. We made the initial simplifying assumptions that: (1) each 60S identified more closely reflects either LN or mature 60S, that is, the number of classes needed to describe all detected targets is two; (2) the nuclear targets more closely resemble the LN 60S and the cytoplasmic targets more closely resemble the mature 60S. We therefore define the prior probability that a randomly selected detected target belongs to a specific population according to the number of targets detected in the nucleus and cytoplasm, respectively (*Figure 2J*, *Figure 2—figure supplement 1A*).:

P(targets in class 1) = P(Nucleus) = 0.11 and,

P(targets in class 2) = P(Cytoplasm) = 0.89.

We used a maximum-likelihood approach to model the $\log_2$(mature / LN 60S 2DTM SNR) values as a mixture of two Gaussians (*Figure 3A*, $R^2$=0.993). The fit suggests a major population with a mean of 0.336, indicating that it more closely reflected the mature 60S, and a smaller population with a mean of –0.026, that slightly more closely reflected the LN 60S (*Figure 3A*). The means of the two fitted populations match the means of the cytoplasmic (0.335) and nuclear (–0.026) populations when considered independently (*Figure 3B*), indicating that our approach effectively separates the nuclear from the cytoplasmic populations. In this case, initializing the prior using the nuclear and cytoplasmic probabilities improved the fit and the agreement with the nuclear and cytoplasmic populations relative to no prior which gave means of –0.038 (class 1) and 0.239 (class 2) (*Figure 3A* vs *Figure 3—figure supplement 1A,B*).

Using the Gaussian distribution model (see Materials and methods), we calculate the probability that a LN and mature 60S-detected target with a given $\log_2$(mature / LN 60S SNR) value derives from class 1 or class 2 via Bayes rule (*Figure 3B–C*). This analysis could easily be extended to cases where more than two templates are used in the search (see Materials and methods). A confidence threshold of 95% assigns ~18% of the nuclear targets and only ~0.2% of the cytoplasmic targets to class 1 (*Figure 3C*). Defining a threshold at 50% classifies ~61% of the nuclear targets as class 1 (nuclear) and 96% of the cytoplasmic targets as class 2 (cytoplasmic) (*Figure 3C*), consistent with the values determined using a threshold $\log_2$ value of 0 (*Figure 2*). The relative probability of each detected 60S belonging to either class can be readily visualized (*Figure 3D*). This shows that the 2DTM SNR ratio can effectively delineate populations of particles in cells based on their relative similarity to similar templates with a specified confidence for each particle assignment.

## Relative similarity to alternate templates reveals population identity

We show above that different 60S populations in the cell can be separated by comparing their relative similarity to alternate templates (*Figure 2*). We also show that these populations can be identified from a mixed population by fitting Gaussians, allowing for the assignment of states with a given probability (*Figure 3*). We sought to investigate factors affecting the assignment of states.

Overfitting the template to background features (noise) could bias the classification of states. The bias in the observed 2DTM SNRs will depend on the degrees of freedom of aligning a template to a detected target. Assuming an error of not more than ±1 pixel in the x,y plane and ±one angular step in the three search angles, we have up to 243 possible ways to align a template to a detected target, allowing the template to partially align itself to noise. This leads to an average apparent noise SNR ($SNR_n$) of $\sqrt{2ln(243)} = 3.31$ (*Grigorieff, 2000*). In this rough calculation, we ignore the defocus as an additional degree of freedom because the signal for neighboring defocus search steps is highly

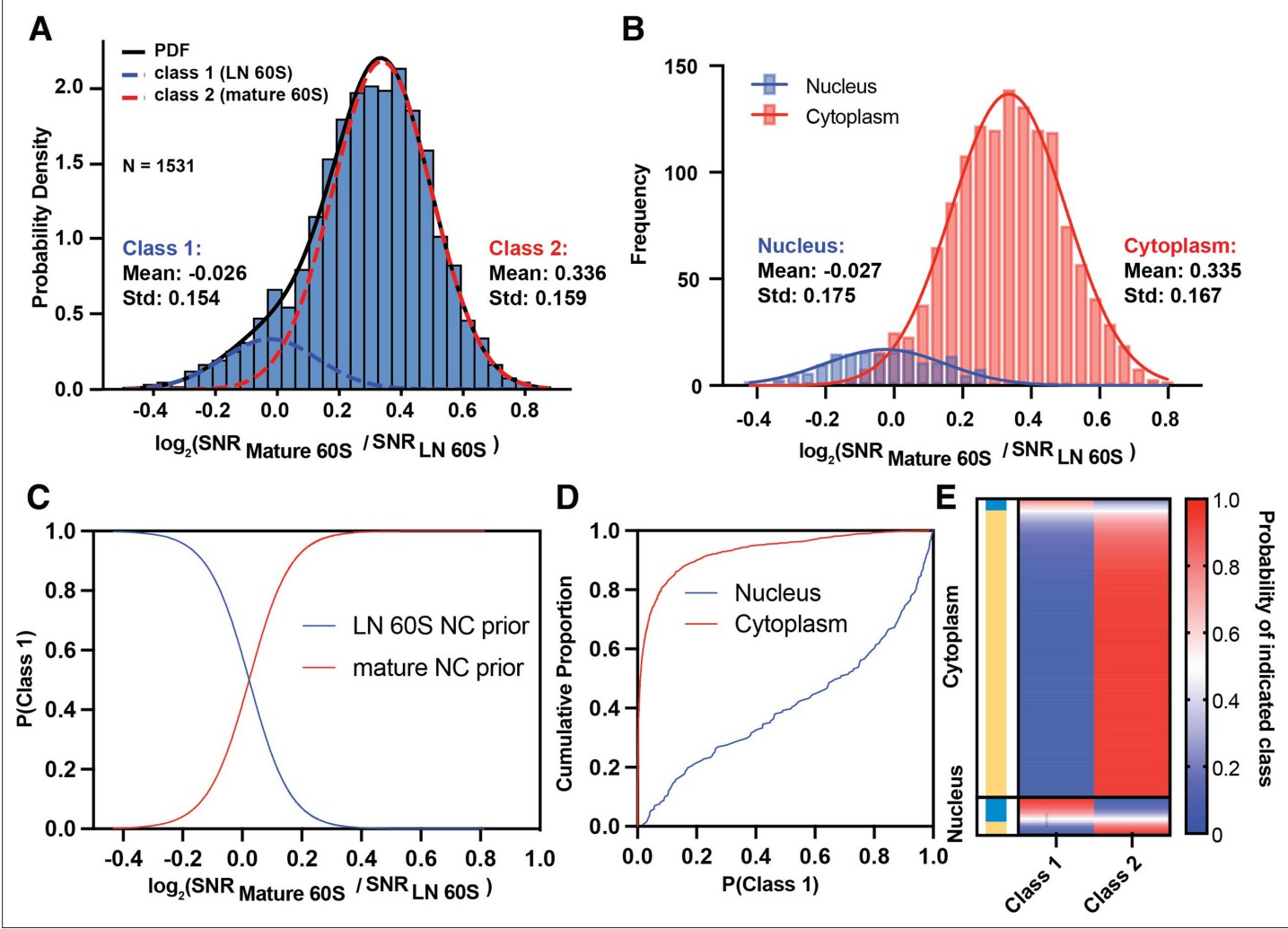

**Figure 3.** Relative probability of detecting mature or LN 60S. (**A**) Histogram showing the distribution of the log₂(mature 60S / LN 60S 2DTM SNR) values for each LN and mature 60S-detected target fit with two Gaussians indicating populations 1 (blue dashed line) and 2 (red dashed line). The black line indicates the sum of the two Gaussians, R²=0.993. (**B**) Histogram showing the log₂(mature 60S / LN 60S 2DTM SNR) values for each nuclear (blue) or cytoplasmic (red) LN and mature 60S-detected target. The two subcellular populations are plotted independently. Solid lines indicate Gaussian fits. (**C**) Line graph showing the probability that a given target belongs to the LN 60S population (blue) line, or mature 60S population (red), as a function of log₂ 2DTM SNR ratio. (**D**) Line graph showing the fraction of nuclear (blue) and cytoplasmic (red) targets classified as LN 60S, at the indicated confidence intervals determined using *Equation 6*. (**E**) Heat map showing the probability of each LN and mature 60S-detected target belonging to either the LN or mature 60S populations. Each row indicates a detected target, and the rows are sorted by their subcellular distribution. The targets assigned to the mature 60S population are indicated in yellow and the targets assigned to the LN 60S population are indicated in blue.

The online version of this article includes the following figure supplement(s) for figure 3:

**Figure supplement 1.** Maximum likelihood fit assuming no prior information.

**Figure supplement 2.** Higher 2DTM SNRs improve classification.

correlated. Since the apparent signal from the aligned noise adds coherently with the true signal of the detected target, the observed $SNR_o = SNR_s + SNR_n$ , with $SNR_s$ the SNR generated by the signal. For the SNR threshold of 7.85 used in our study, this means that $SNR_s$ might only be 4.54, about 40% lower, a substantial difference. For higher observed SNRs, the percentage noise bias will be smaller. When several similar templates are used, as in the present study, the average noise bias in the observed 2DTM SNRs will approximately be the same, substantially reducing its effect on the SNR ratios and classification results. For two templates, the SNR ratio is given by

$$SNR_{o,1}/SNR_{o,2} = (SNR_{s,1} + SNR_{n,1})/(SNR_{s,2} + SNR_{n,2}) \qquad (1)$$

which will be approximately invariant with $SNR_n$ for $SNR_s \gg SNR_n$, that is, independent of the background level in the image. For SNR values closer to the detection limit, this ratio will exhibit a larger variance and biased towards 1 ($\log_2$ values will be biased towards 0), making it more difficult to unambiguously assign target identity.

Accordingly, we note above that the $\log_2$(mature / LN 60S 2DTM SNR) values of the cytoplasmic 60S population could be fit by a single Gaussian and clearly differentiated from the nuclear 60S population

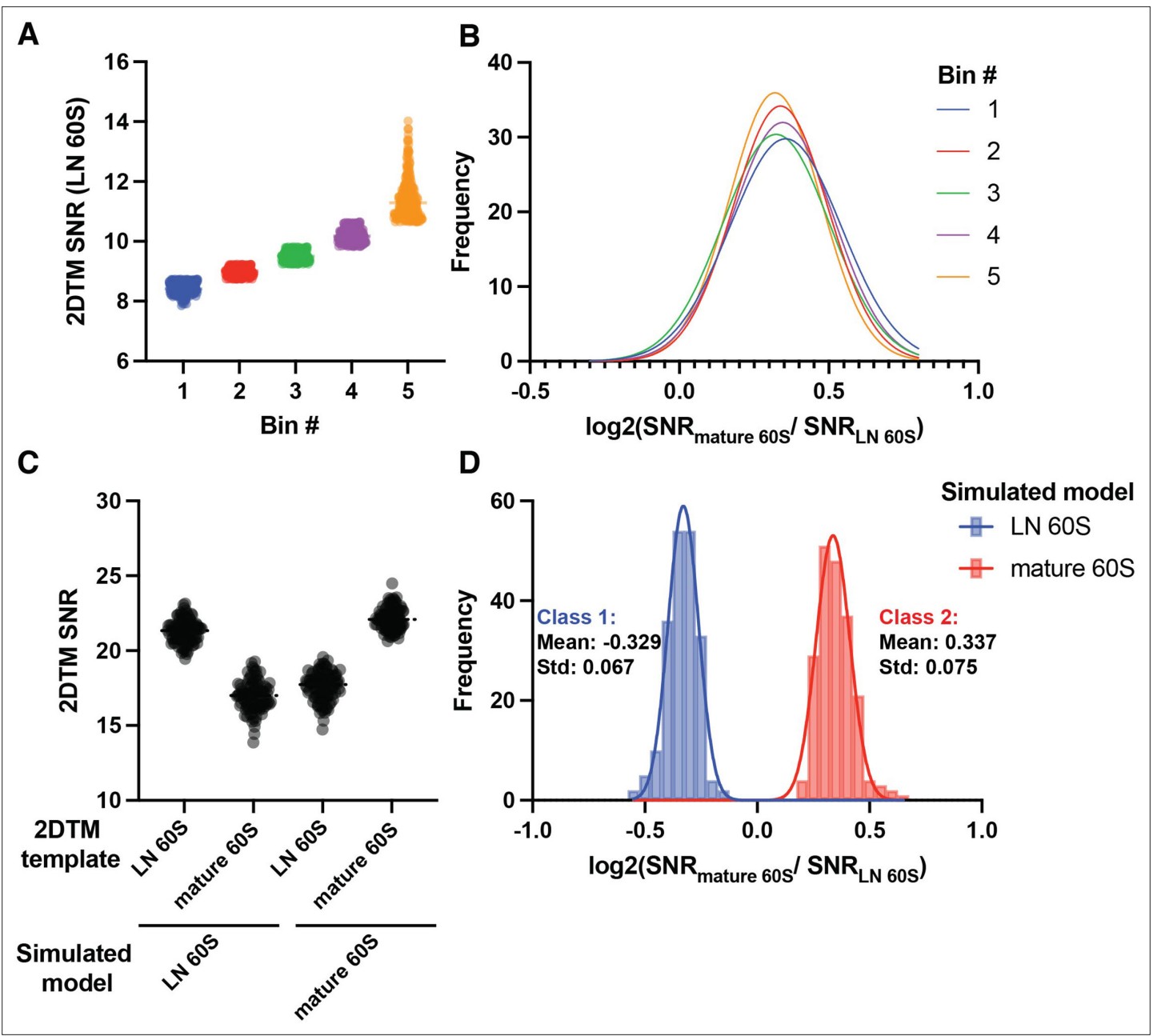

**Figure 4.** Simulations reveal identity of cytoplasmic 60S population. (**A**) Scatter dot plot showing the 2DTM SNR of all cytoplasmic LN 60S and mature 60S-detected targets, grouped into five bins of 272 targets by the 2DTM SNR of the LN 60S-detected target, each representing 20% of the cytoplasmic targets. (**B**) Gaussian fits to histograms of the $\log_2$(mature 60S / LN 60S 2DTM SNR) representing the bins shown in (A), where bin 1 represents the bottom 20%, and bin 5 represents the top 20% of cytoplasmic targets by their LN 60S 2DTM SNR. (**C**) Scatterplot showing the 2DTM SNRs using the indicated templates to search simulated images of LN 60S (left) or mature 60S (right). (**D**) Histogram of the $\log_2$(mature 60S / LN 60S 2DTM SNR) values resulting from particle-wise comparison of the values in the simulation shown in (C). Solid lines indicate Gaussian fits for two distinct populations.

(*Figure 3*). This is striking because the 60S detections were combined from multiple images with differing SNRs. Moreover, the mean $\log_2$(2DTM SNR ratio) of the top 20% of targets was only ~10% different from the bottom 20% of targets based on their LN 60S 2DTM SNR (*Figure 4A–B*), consistent with the ratio of SNRs not depending strongly on the SNR values.

To test this further we generated simulated images of LN and mature 60S in ice alone, without overlapping proteins and other density (*Figure 4C*). We found that the mean $\log_2$ value of the mature 60S population in the simulation was 0.337 (*Figure 4D*), which agrees with the observed mean $\log_2$(mature / LN 60S 2DTM SNR) value of the cytoplasmic population (p=0.867, unpaired t-test). This result indicates that the cytoplasmic population closely matches the mature 60S template.

Our simulation suggests that, above a certain 2DTM SNR the mean $\log_2$ values can be predicted solely based on the templates, and a deviation from this ratio indicates that the detected targets have a significant mismatch with either template. In contrast to the cytoplasmic population, the nuclear population deviates from the predicted $\log_2$(mature / LN 60S 2DTM SNR) (*Figure 4D*). This indicates that targets in this population deviate significantly from both the mature 60S and the LN 60S templates. This provides further evidence that the nuclear population likely contains a more complex mix of maturation states (see below).

## Ribosome biogenesis factors differentiate nuclear from cytoplasmic 60S

The nuclear and cytoplasmic 60S populations differ with respect to their relative similarity to the LN and mature 60S templates (*Figure 2*). Classification based on their $\log_2$ 2DTM SNR ratios (*Figure 3*) and comparison with predicted ratios for the mature and LN 60S templates (*Figure 4*) identified the vast majority of the cytoplasmic targets as mature 60S. However, the mean $\log_2$ values of the nuclear 60S population were close to 0, indicating that the nuclear 60S population is distinct from both the LN and mature 60S templates. To investigate this further, we assessed the features of the two templates that distinguish the nuclear from the cytoplasmic populations.

Most of the mass difference between the LN and mature 60S templates results from proteins in the LN 60S that are absent in the mature 60S (*Figure 5A–C*). Notable exceptions are the proteins on the P-stalk which are present only on the mature 60S (*Figure 5A–C*, *Figure 3A*). Additionally, several rRNA helices on the intersubunit interface are in different conformations, specifically the L1 stalk, helix 38 and helix 89, which undergo conformational changes during maturation (*Figure 5C*). To identify which of these features distinguish nuclear from cytoplasmic 60S, we investigated the relative dependence of the 2DTM SNRs on the rRNA and proteins of the LN 60S template. We generated truncated LN 60S templates containing either rRNA or protein only and calculated the change in the 2DTM SNR for each template at each target relative to the full-length template (*Figure 5D*). The rRNA contributed 1.5 and 1.8-fold more to the 2DTM SNR of the nuclear and cytoplasmic targets, respectively, despite comprising only 1.25-fold more of the template mass (1004 and 800 kDa, respectively), than the proteins (*Figure 5D*). Indeed, 60% of the cytoplasmic targets and 34% of the nuclear targets were no longer significant when searching with the proteins alone. Comparing the nuclear and cytoplasmic populations shows that the 2DTM SNR of the LN 60S-detected cytoplasmic targets is less affected by the removal of the LN 60S proteins and more strongly affected by the removal of the rRNA from the template density (*Figure 5D*). This shows that the LN 60S proteins contribute positively to the SNR of the nuclear targets and negatively to the cytoplasmic targets and therefore differentiate the nuclear from the cytoplasmic 60S population.

Since the LN 60S represents a late intermediate of 60S maturation in which the rRNA is almost fully folded, RBF proteins on the LN 60S account for most of the difference with the mature 60S by mass (*Figure 5A–D*). To confirm that the SNR difference of nuclear LN 60S-detected targets and cytoplasmic mature 60S-detected targets is primarily due to the RBF proteins, we removed the RBFs from the LN 60S template and recalculated the SNR for each target. The removal increased the 2DTM SNR of the cytoplasmic targets, while decreasing the 2DTM SNR of the nuclear targets (*Figure 5E*), making the SNR values more similar. This is consistent with the nuclear population having these RBFs and the cytoplasmic population lacking the RBFs. We conclude that the differentiation of detected targets using the observed 2DTM SNRs reflects biologically relevant differences between them.

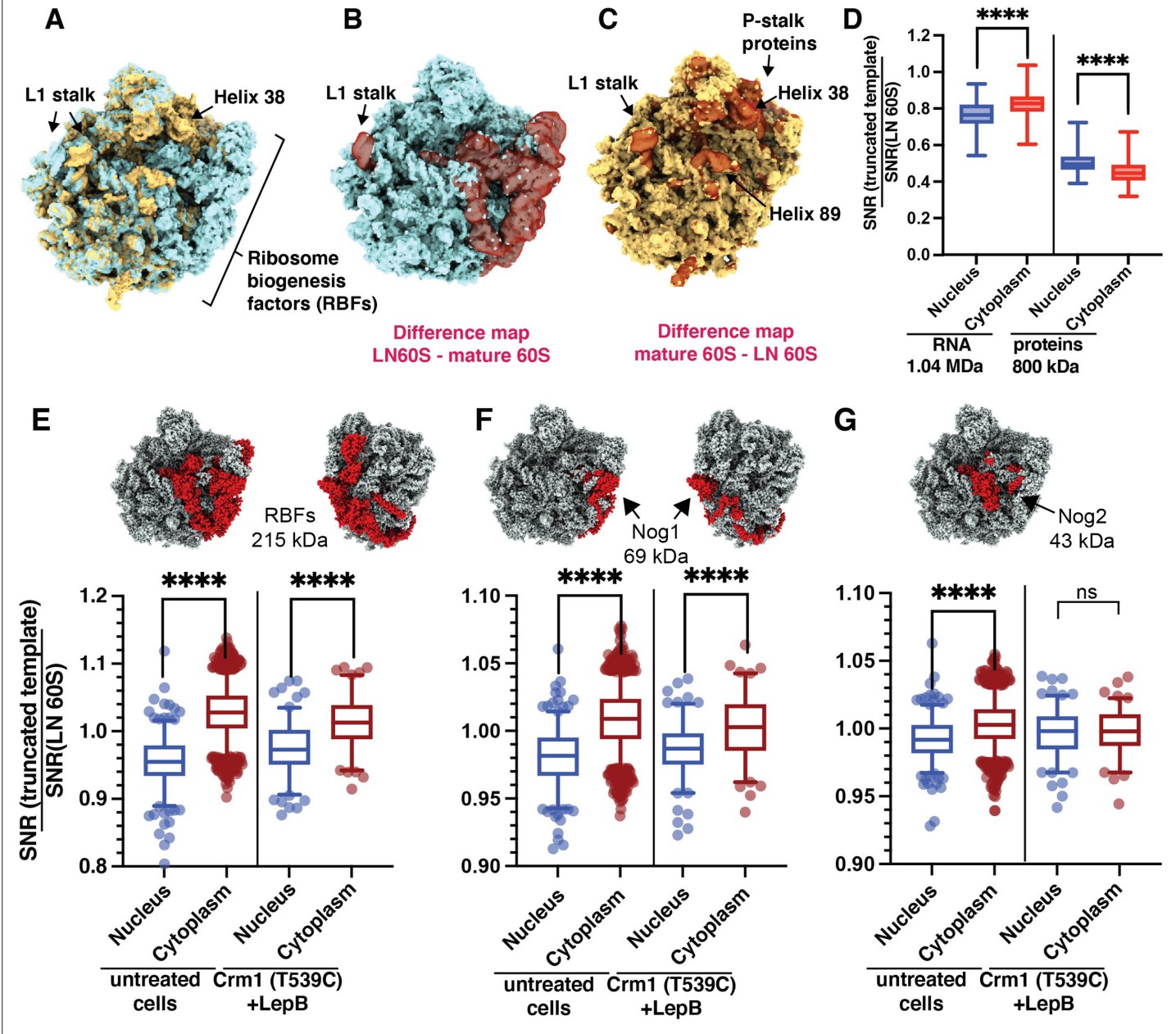

**Figure 5.** Classification of cytoplasmic mature 60S and nuclear pre-60S by 2DTM corresponds with biologically relevant differences in the templates. (**A**) The LN 60S (blue) and mature 60S (yellow) 2DTM templates aligned in UCSF Chimera. (**B**) LN 60S with difference map calculated using UCSF Chimera showing the density in the LN 60S template that is not present in the mature 60S template (red, transparent). (**C**) As in (B), showing the mature 60S with density that is not in common with the LN 60S template (red, transparent). (**D**) Boxplots showing the change in 2DTM SNR when only RNA (left) or protein (right) components of the LN 60S template are included, relative to the full-length template for each significant target. The targets are grouped by their subcellular localization. (**E**) Upper: LN 60S template with all ribosome biogenesis factors (RBFs) indicated in red. Lower: Boxplot showing the change in the 2DTM SNR of the nuclear (blue) and cytoplasmic (red) targets when all RBFs are removed, relative to the full-length LN 60S template in untreated cells, and when Crm1-mediated nuclear export is inhibited by treating Crm1 (T539C) cells with Leptomycin B (LepB). Box width indicates the interquartile range, the central line indicates the median and the whiskers indicate the range of 95% of the targets. (**F**) As in (E), for RBF Nog1. (**G**) As in (E), for RBF Nog2. ****: p<0.0001, ns: not significant (p>0.05).

The online version of this article includes the following figure supplement(s) for figure 5:

**Figure supplement 1.** Detection of LN and mature 60S in the nuclear and cytoplasm in Crm1 (T539C) cells treated with Leptomycin B.

## Nog2 lacking intermediates accumulate after inhibition of nuclear export

The two largest RBFs on the LN 60S are Nog1 and Nog2, together accounting for ~50% of the RBF mass (*Figure 5F and G*). During 60S maturation, Nog2 removal is required to permit binding of the nuclear export adaptor Nmd3 and Crm1-dependent export, and therefore Nog2 removal precedes nuclear export (*Ho et al., 2000*; *Matsuo et al., 2014*). In contrast, Nog1 is removed only upon export to the cytoplasm (*Pertschy et al., 2007*). In cells with active nuclear export, we find that removal of either Nog1 or Nog2 differentiates the nuclear from the cytoplasmic populations (*Figure 5F and G*, untreated cells). As a further test of differentiating different targets by 2DTM, we inhibited Crm1 mediated export by treating Leptomycin B (LepB) sensitive Crm1 (T539C) cells (*Neville and Rosbash, 1999*) with LepB and located 60S targets with LN 60S and mature 60S templates in eight images of FIB-milled lamellae. To assess the relative occupancy of Nog1 and Nog2 after Crm1 inhibition, we measured the change in 2DTM SNR after removal of all RBFs, and Nog1 or Nog2 alone. Consistent with LepB inhibiting export of pre-60S from the nucleus, we detected a higher density of pre-60S in the nucleus than in cells with active Crm1 (*Figure 5—figure supplement 1A*, *Figure 6F*). When nuclear export is inhibited, all RBFs (*Figure 5E*) and Nog1 alone (*Figure 5F*) still differentiate the nuclear from the cytoplasmic populations. In contrast, the occupancy of Nog2 is no longer significantly different between the nuclear and cytoplasmic populations (*Figure 5G*). This is consistent with a model in which, when Crm1-mediated export is active, nuclear intermediates are rapidly exported after removal of Nog2, depleting the Nog2-lacking population from the nucleus. In the presence of a Crm1-inhibitor, the late, export competent nuclear intermediate lacking Nog2 can no longer be exported and therefore accumulates. Since Nog1 is only removed after export, inhibition of export did not change the occupancy of Nog1 on the maturing 60S. This demonstrates that comparing 2DTM SNRs is sufficiently sensitive to assess the changes in the occupancy of individual proteins on 60S complexes in situ.

## Classification of nuclear pre-60S intermediates

Ribosome biogenesis is a highly efficient molecular assembly line, and multiple intermediate states co-exist in the cell (*Warner, 1999*). Therefore, the nuclear population of pre-60S is unlikely to represent a single intermediate population. Accordingly, the distribution of the mature 60S / LN 60S SNR ratios of nuclear mature and LN 60S-detected targets fits a single Gaussian more poorly than the cytoplasmic targets (*Figure 2J*), and the mean $\log_2$ value is close to 0, suggesting that additional nuclear populations, distinct from either template, were identified with both 60S templates. To test this prediction and investigate the nuclear pre-60S population further, we generated a third template corresponding to an earlier nuclear intermediate (EN 60S). EN 60S (PDB: 3JCT) retains internally transcribed spacer RNA 2 (ITS2) and associated proteins and has 5S rRNP in a premature state rotated 180° relative to the LN and mature 60S (*Figure 6A*; *Wu et al., 2016*). We identified 679 significant EN 60S-detected targets of which 545 (~80%) were also identified with the LN 60S template, and 489 (72%) were also identified with the mature 60S. All of the 489 EN 60S-detected targets identified with the mature 60S were also identified with the LN 60S (*Figure 6A*). 289 (43%) of the EN 60S-detected targets localized to regions of the images corresponding to the nucleus, similar to the 268 nuclear LN 60S-detected targets, while only 390 were cytoplasmic, >3 fold fewer than located with the LN 60S template, consistent with the EN 60S representing a less mature nuclear intermediate (*Figure 6B*). The number and localization of targets identified with 2DTM is consistent with their sequence in the maturation pathway, progressing from EN 60S to LN 60S in the nucleus to mature 60S in the cytoplasm.

Cross-detection of targets by different templates can be used to detect heterogeneity in target populations. When examining the SNR ratios of targets identified by both EN and LN 60S, the cytoplasmic targets display a distribution that is consistent with a single population that more closely resemble the LN 60S template (*Figure 6—figure supplement 1C*, **red**). The distribution of nuclear targets, however, was consistent with at least two populations (*Figure 6—figure supplement 1C*, **blue**), each of which is distinct from the cytoplasmic population. A similar pattern was observed when comparing the relative similarity of each identified 60S to the EN 60S or mature 60S (*Figure 6—figure supplement 1D*). This indicated the presence of at least two nuclear populations that differ with respect to their relative similarity to the EN, LN and mature 60S templates.

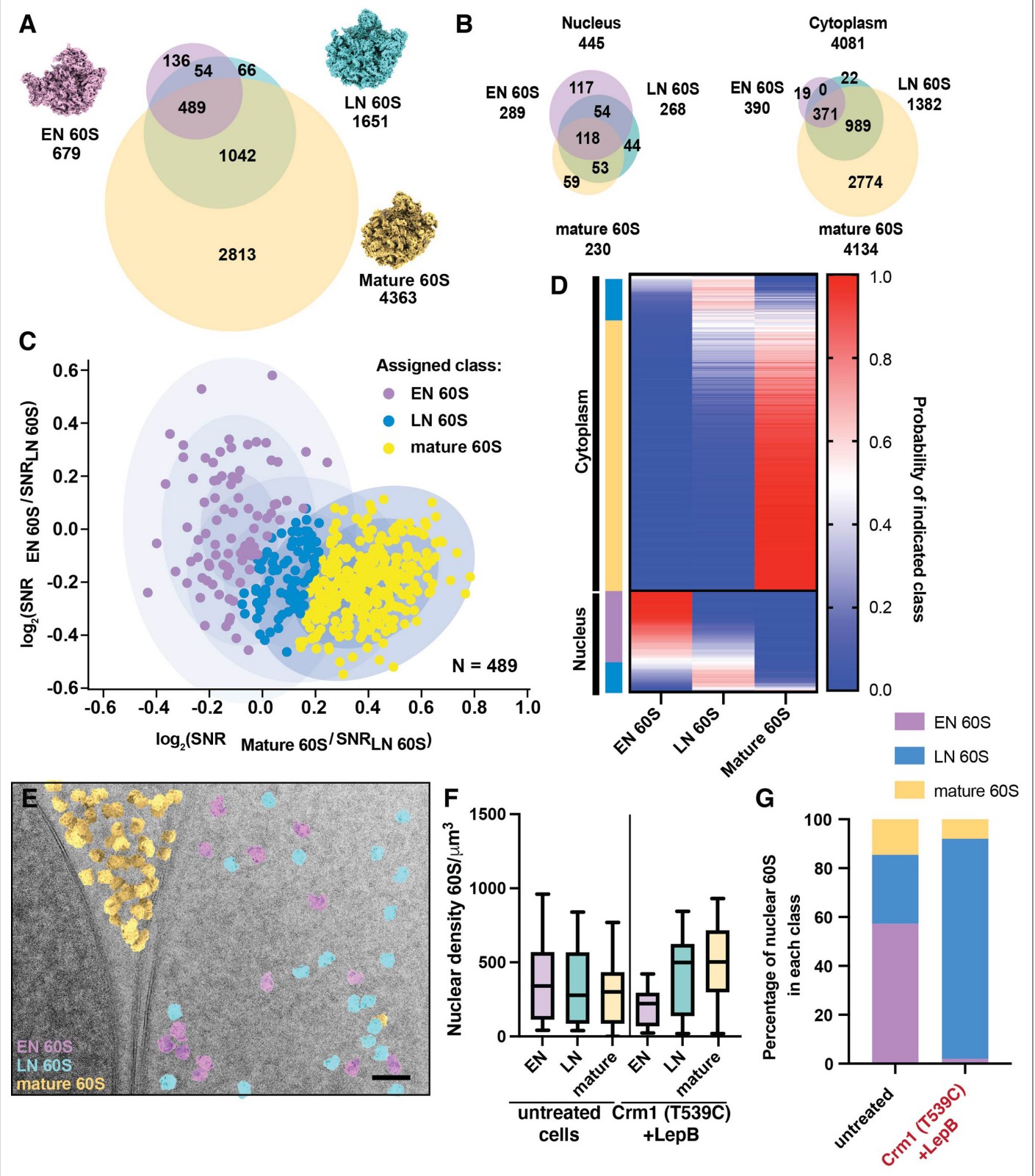

**Figure 6.** Classification of nuclear targets by relative similarity to early or late nuclear intermediates. (**A**) Venn diagram showing the number of significant targets detected in 2DTM searches with the indicated templates. Overlap indicates targets identified in two or more searches. (**B**) Venn diagrams showing the number of significant targets detected in 2DTM searches with the indicated templates in the nucleus (left) and cytoplasm (right). (**C**) Scatterplot showing the log$_2$(EN 60S / LN 60S 2DTM SNR) values relative to the log$_2$(mature 60S / LN 60S 2DTM SNR) values for each EN, LN and

*Figure 6 continued on next page*

*Figure 6 continued*

mature 60S-detected target. Ellipses indicate the fits of three Gaussians and each concentric ellipse indicates one standard deviation from the mean. Each target is colored according to its most likely class membership. (D) Heat map showing the probability of each of the targets examined in (C) belonging to one of the populations, EN, LN or mature 60S. Targets are grouped by their subcellular localization, followed by their classification as EN 60S (purple), LN 60S (light blue), or mature 60S (yellow). (E) Cryo-EM micrograph of the yeast nuclear periphery from a FIB-milled lamella shown in in *Figure 1*, displaying the results of 2DTM searches, colored by their classification as mature 60S (yellow), LN 60S (blue) or EN 60S (purple) based on their relative 2DTM SNRs. (F) Boxplot showing the nuclear density of EN 60S (purple), LN 60S (light blue), and mature 60S (yellow) detected targets before classification in the indicated cells. (G) Bar chart showing the proportion of the LN 60S-detected targets in the indicated cells that are classified as LN 60S (blue), mature 60S (yellow), or EN 60S (purple).

The online version of this article includes the following figure supplement(s) for figure 6:

**Figure supplement 1.** Relative similarity to EN, LN and mature 60S distinguishes the nuclear and cytoplasmic 60S populations.

**Figure supplement 2.** Additional image showing results of classification using EN, LN and mature 60S.

We next sought to classify the EN, LN and mature 60S-detected targets based on their relative similarity to the three 60S templates. For each target we calculated the $\log_2$(mature 60S / LN 60S SNR) and $\log_2$(EN 60S / LN 60S SNR) values. We used these values to classify each target based on the relative similarity to the three templates using the maximum-likelihood approach discussed above (*Figure 6C*). We found that, consistent with their expected subcellular distributions, targets assigned to the mature 60S population represented 315 (85%) of the cytoplasmic targets and only 1 (<1%) of the nuclear targets detected by all three templates (*Figure 6D*). In contrast, the EN 60S population represents 83 (70%) of the nuclear population and only 4 (~1%) of the cytoplasmic population detected with all three templates (*Figure 6D*). The LN 60S population was roughly evenly distributed between the nucleus and the cytoplasm, consistent with this structure representing a late maturation intermediate (*Figure 6D*). These results can be visualized by annotating each detected 60S target with the template it most closely resembles (*Figure 6E*, *Figure 6—figure supplement 2*).

The NE provides a convenient visual control for the classification of targets as LN / EN 60S or mature 60S (e.g., *Figure 1*). However, there are no clear features in the nucleoplasm that would enable visual separation of different populations of nuclear intermediates and thereby confirm their classification. To validate our classification of the nuclear pre-60S populations, we identified conditions wherein the relative occupancy of the two states would be expected to change. We show above that inhibiting Crm1-mediated export results in accumulation of nuclear intermediates that lack Nog2 (*Figure 5*). In cells with active Crm1, there are similar numbers of nuclear EN, LN and mature 60S detected targets (*Figure 5F*) and 57% of the nuclear 60S targets are assigned to the EN 60S population (*Figure 6G*). After inhibition of Crm1-mediated export, the EN 60S population is mostly depleted, while the density of nuclear LN and mature 60S-detected targets increases (*Figure 6F*), and >90% of targets are assigned to the LN 60S population (*Figure 6G*). This confirms that 2DTM SNR ratios can be used to effectively classify mixed populations of particles in cells.

## Discussion

The immense potential for cryo-EM to reveal the molecular detail of biological processes in cells is currently largely unrealized. One of the major bottlenecks is the lack of reliable, quantitative methods to locate and characterize molecules in cells. Here, we describe the application of 2DTM to in situ particle classification. By considering the relative 2DTM SNRs of alternate templates at a single location and orientation, we separate 60S precursors in the nucleus from mature 60S in the cytoplasm. We also show that a maximum likelihood approach effectively classifies a mixed population of nuclear pre-60S into at least two maturation states with a specified confidence for each particle. We show that 2DTM can be used to probe the composition of complexes in situ by modifying 2DTM templates. In this study we extend the utility of 2DTM beyond a binary indicator of detection to provide a quantitative assessment of particle identity.

## 2DTM enables specific molecule localization in the dense interior of cells

Cryo-FIB milled eukaryotic cells are sufficiently well preserved to allow imaging with cryo-ET (*Mahamid et al., 2016*) and subtomogram averaging to yield 3D reconstructions at resolutions of >~12 Å, e.g. (*Schaffer et al., 2019*). However, before the present work it was unclear if the milling preserves the high-resolution signal in these samples sufficiently well to allow for particle detection with 2DTM. Our results clearly show that FIB-milling is compatible with molecule localization by 2DTM. This expands the application of 2DTM to previously inaccessible cell types and further demonstrates the utility of 2DTM for in situ structural biology.

In many images, 60S subunits detected by 2DTM also generate low-resolution contrast in the cytoplasm that is readily visible (*Figure 1B*, yellow arrows). In the nucleoplasm, the similar density of RNA and DNA impedes the visual identification of all but a few pre-60S (*Figure 1B*, blue arrows). However, the reduced low-resolution contrast does not preclude effective detection of pre-60S with 2DTM. This is in contrast to particle localization in tomograms, wherein detection depends more strongly on low-resolution contrast and recognizable shapes.

The ability to distinguish particles in crowded molecular environments is a major advantage of 2DTM relative to cryo-ET, which currently suffers from strong attenuation of high-resolution signal (large B-factors) in the raw tomogram (*Schur et al., 2016*). 2DTM may enable localization of molecules in other dense environments such as liquid-liquid phase separated granules, which remains challenging for cryo-ET despite success in some cases (*Erdmann et al., 2021*). Our results confirm that 2DTM is an effective method to localize molecules in dense regions of the cell even when the molecules cannot be distinguished by eye.

## 2DTM enables single particle classification in situ

In previous work we and others have demonstrated that, when comparing populations of molecules, the average 2DTM SNRs reflect the relative similarity of different templates to the target populations (*Lucas et al., 2021*; *Rickgauer et al., 2020*). In this study, we extend this observation to show that the relative 2DTM SNRs of aligned templates *at a specific location and orientation* can be used to calculate the relative probabilities of a target belonging to a specific particle population.

Of the nuclear targets identified with the mature 60S, ~50% were also detected with the EN 60S, all of which were also detected with the LN 60S (*Figure 6B*). When calculating the relative similarity to the three 60S templates, the EN 60S and mature 60S population were clearly distinct, with mean 2DTM SNR ratios more than three standard deviations apart (*Figure 6C*). The maximum likelihood estimation of Gaussian distributions enables quantitative classification even when particle populations are less distinct, by yielding relative probabilities for each detected target belonging to one of a given number of populations (e.g. *Figures 3 and 6*). The observed shift in the nuclear population towards a more mature intermediate after inhibition of nuclear export provides a biological control that validates our assignment of states.

In this study, we effectively classify at least three populations of 60S maturation states from a population of <500 molecules (*Figure 6*). This means that given sufficient abundance of the target, it will be possible to distinguish populations based on data from a single image (*Figure 2—figure supplement 1D*). This contrasts with more traditional (reference-free) methods used to classify subtomograms and single particles, which require hundreds to thousands of particles to generate the class averages needed for particle assignment. 2DTM allows single molecule classification from fewer images, and therefore enables more information to be extracted from images collected from cells and purified samples (single-particle cryo-EM).

## Confidence metric for single particle classification in situ

Calculating the confidence in class assignment of individual particles will aid interpretation of the results of 2DTM in situ. One major difference between in situ cryo-EM and single-particle cryo-EM is the type of biological information that is obtained. In single-particle cryo-EM, the goal is to generate high-resolution maps and establish the arrangement of atoms within a complex in different functional states, and to use this information to discern its molecular mechanism. In this case, B-factors and other metrics can be used to indicate uncertainty about an atomic coordinate, which aids interpretation of the model built into the map. In the cell, each individual instance of a complex may be in a different

context relative to other similar molecules. For example, particles might be in different subcellular compartments such as the nucleus or cytoplasm or, as a more extreme example, a single particle within a nuclear pore exists in a very different context than particles in the nucleoplasm. For structural cell biology applications, therefore, it is useful to define a metric to establish the confidence of single particle classification. In this study, we show that a maximum likelihood approach using Gaussian fits to $\log_2$ 2DTM SNR ratios of alternate templates at a specific subcellular location and orientation can be used to calculate the relative probability of a single particle deriving from one of a given number of classes. This provides a quantitative metric to establish confidence in the assignment of single particles that will aid in the biological interpretation of cellular cryo-EM maps.

## 2DTM templates as computational molecular probes

A major challenge in biological cryo-EM is the retrieval of detailed structural information of inherently flexible and heterogeneous macromolecules from noisy images collected at low dose to limit radiation damage. In single particle cryo-EM, this problem is addressed by averaging images of thousands of purified molecules to identify different structural states at high resolution. By averaging images of many identical copies of a particle, novel structures can be discovered, and this is a clear strength of this approach. However, since most complexes have a low abundance in the cell, the utility of this approach for in situ structural biology is limited to all but the most abundant complexes.

2DTM presents an alternate approach to using the signal in noisy images to gain insight into the structural states of molecules. In this approach, a noise-free template represents a hypothesis that a particle of a given conformational and compositional state is present in the image, and this hypothesis can be tested by searching the image with the template, independent of how many particles the image contains. We demonstrate that by generating modified templates representing different hypotheses, we can directly assess the compositional and conformational states of ribosomal subunits in cells.

Provided the templates have similar molecular mass and shape and are aligned with each other, probing with multiple templates requires only a single initial exhaustive search with one of the templates. This can be followed by a simple evaluation of the cross-correlation coefficient for each additional template at locations and orientations of the detected targets in the initial search (*Figure 5*), thereby avoiding time-consuming searches for all templates. In future studies, this approach could be extended to assess the relative similarity of a target with respect to a library of alternate structures. Alternate templates could be generated in multiple ways, depending on the biological hypothesis being tested. To reveal compositional heterogeneity in situ, alternate structures could be generated that lack specific subunits of interest as shown in *Figure 5*. Additionally, to interrogate in situ conformational heterogeneity, templates could be generated from time points of molecular dynamics simulations.

Our finding that, unlike the 2DTM SNR, the mean $\log_2$ of SNR ratios of a population is not strongly dependent on image SNR, allowed us to determine whether a given model matches the data. This approach could vastly streamline validation of models from in vitro or in silico experiments using relatively few images. Hypothetically, if models covering the full conformational and compositional space could be generated, the likelihood of a particular structure could be calculated for individual molecules.

## Overfitting and classification

Assuming additive Gaussian noise, we estimated that overfitting contributes substantially to the variance in the observed 2DTM SNRs from an individual search. The effect of overfitting is proportionally greater for lower 2DTM SNRs and less for higher 2DTM SNRs, making higher 2DTM SNRs more reliable. However, while overfitting can also affect the classification of targets using their $\log_2$(SNR ratios), the effect is substantially less. Overfitting will introduce some bias in the $\log_2$ values that may affect classification. Misclassification may occur when the signal SNRs are similar. At high 2DTM SNRs, the proportional difference in the $\log_2$ values will be small and are unlikely to affect the classification. At low SNRs close to the threshold overfitting will have a greater effect, making classification of this population more challenging. However, collectively these factors will be captured by modelling the populations with Gaussians. Using a Gaussian model to derive the probability of class membership, rather than using a single threshold, we account for the bias in the $\log_2$ values.

## Addressing additional potential sources of error

In our study, we used the physical separation of nuclear and cytoplasmic 60S populations to develop and test in situ classification of targets by 2DTM. We found that there are several requirements to permit classification of related molecules by 2DTM. First, the molecular models must be aligned relative to one another resulting in a correlation peak at the same pixel in the image. Comparing SNR values resulting from global searches with different templates may be lowered by imperfect, off-grid rotational matches, potentially affecting 2DTM SNR ratios and hence, target classification. Differences in model quality may also affect the 2DTM SNR ratios, masking other differences of interest. In this study, the mature 60S template was generated using the atomic coordinates of the large subunit of the ribosome built into a map with an overall resolution of 3.1 Å (PDB: 6Q8Y) (*Tesina et al., 2019*). The large subunit of the ribosome is structurally less variable than the small subunit and local resolution estimates suggest that parts of the LSU map extend to ~2.5 Å (*Tesina et al., 2019*). The maps used to build the EN 60S and LN 60S subunits were reconstructed at 3.08 Å and at 3.5 Å resolution, respectively. The accuracy of the atomic coordinates of a model will depend on the resolution of the underlying density map. Moreover, the greater number of mature ribosome structures, relative to maturation intermediate structures, may provide more confidence in the atomic coordinates of the mature 60S. The putative higher accuracy of the mature 60S model, relative to the LN 60S model, may partially explain the closer agreement of predicted and observed SNR ratios of the cytoplasmic population, compared to the nuclear population. We expect that more accurate coordinates will result in higher 2DTM SNR values, which may affect target classification.

False positive detections could skew the classification results by introducing false additional populations. In our analysis we use a Gaussian noise model to apply a threshold that permits a single false positive per image (*Lucas et al., 2021*; *Rickgauer et al., 2017*), although this is likely to be an overestimate (*Lucas et al., 2021*). Since 28 images were searched, we expect up to 28 false positives of the 4363 mature 60S-detected targets (0.6%). Since we limited our analysis to targets detected by multiple templates, the false positive rate within these targets will be even lower. Assuming the worst-case scenario of 28 false-positives in the 489 targets detected by all three templates, this would represent a false-positive rate of 6% for the smallest class of EN 60S-detected targets. We therefore do not expect that false positives play a major role in the assignment of 60S states. For low-abundance complexes with a detection of only a few tens of targets, the relative proportion of false positives would be higher, and the detection threshold would therefore need to be adjusted to lower the false-positive rate. Future improvements to the 2DTM algorithm, such as integration of correlation peaks over all search parameters and peak profile fitting could reduce template overfitting and improve the separation of the signal and noise distributions (*McDonough and Whalen, 1995*) and the identification and classification of smaller or less abundant complexes.

The classification of structurally similar targets could be further improved by identifying and controlling the factors that affect the distribution of observed 2DTM SNR ratios for a given set of templates. Ideally, the mean ratio of SNR values for a set of templates and given target depends only on the structural differences between the templates, while the distribution of observed ratios is solely a function of the noise and background in the images and target orientation. However, factors that contribute to loss of signal such as sample thickness, radiation or FIB-milling damage, beam-induced motion, charging and movie frame alignment errors due to sample deformation all result in loss of high-resolution signal, making the 2DTM SNR ratios less sensitive to structural differences in the templates, biasing their $\log_2$ values towards 0 and increasing the variation due to increased relative noise resulting in different degrees of overfitting. The cell is a highly heterogeneous environment, and systematic local differences in noise, such as between subcellular compartments may also affect classification. In this case, context-specific biological controls are needed to validate the assignment of states. Further research is required to account for these factors and reduce the variance in 2DTM SNR ratios, thereby enabling classification of targets with smaller structural differences.

## Additional intermediate populations

In the present study, we only considered three alternate 60S templates. We note that the Gaussian fits to the 2DTM SNRs of mature 60S and LN 60S-detected nuclear targets is imperfect, potentially indicating additional pre-60S populations (*Figure 2—figure supplement 1C*). Further examination of the observed 2DTM SNR ratios revealed the presence of at least one additional pre-60S population

(*Figure 6*). The observed shift towards the more mature LN 60S intermediate when nuclear export was inhibited is evidence for detection of a biologically relevant pre-60S population (*Figure 6G*). We also observed a small population of cytoplasmic 60S targets with higher SNR values against the LN 60S template than against the mature 60S (*Figure 6D*). 60S maturation intermediates exit the nucleus in an immature form and complete maturation in the cytoplasm. Whether the cytoplasmic 60S with higher SNR values against the LN 60S template represent cytoplasmic intermediates or reflect the limits of our classification strategy requires further investigation. Future work using additional templates representing other intermediates of 60S maturation will reveal further details about the spatiotemporal organization of pre-60S intermediates in cells.

In this study, we identified an EN 60S population of nuclear 60S with the 5S rRNP in a premature state rotated 180° relative to the mature 60S, consistent with in vitro determined structures (*Leidig et al., 2014*). The presence of this complex during maturation in vivo has been difficult to establish. Our observation that this population accounts for more than half of the 60S identified in the nucleus argues that this is an on-pathway assembly intermediate. We also identified a nuclear LN 60S population. This population reflects a late intermediate that has already undergone 5S rotation and ITS2 removal, implying a temporal lag after 5S rotation and/or ITS2 removal, and subsequent export from the nucleus. To test these possibilities more thoroughly, future studies establishing the flux through the assembly pathway are needed. By freezing cells at different time points after inhibition of specific maturation steps, 2DTM could be used to study the kinetics of assembly and the flux through the assembly pathway.

## Materials and methods

### Key resources table

| Reagent type (species) or resource | Designation | Source or reference | Identifiers | Additional information |
|---|---|---|---|---|
| Strain, strain background (*Sacchromyces cerevisiae*) | BY4741 | ATCC | S288C | |
| Strain (*Sacchromyces cerevisiae*) | Crm1 (T539C) | *Neville and Rosbash, 1999* | MNY8 | |
| Software, algorithm | cisTEM | *Grant et al., 2018*; *Lucas et al., 2021* | | https://cistem.org/development |

### Yeast cell culture and plunge freezing

*Saccharomyces cerevisiae* strains BY4741 (ATCC), or Crm1 (T539C) (MNY8 *Neville and Rosbash, 1999* a gift from Michael Rosbash, Brandeis) colonies were inoculated in 20 mL of YPD, diluted 1/5 and grown overnight at 30 °C to an $OD_{600}$ of ~0.5–1. The cells were then diluted to 10,000 cells/mL and 3 µL applied to a 2/1 or 2/2 Quantifoil 200 mesh Cu grid, allowed to rest for 15 s, back-side blotted for 8 s at 27 °C, 95% humidity followed by plunge freezing in liquid ethane at –184 °C using a Leica EM GP2 plunger. Frozen grids were stored in liquid nitrogen until FIB-milled. When indicated Crm1 (T539C) cells were additionally incubated at 30 °C with shaking in the presence of 200 nM Leptomycin B (Cell Signaling Technologies) for 30 min before applying to grids and plunge freezing.

### FIB milling

Grids were transferred to an Aquilos cryo-FIB SEM, sputter coated with metallic Pt for 15 s then coated with organo-Pt for 10 s and milled in a series of sequential milling steps using a 30kV Ga +beam using the following protocol: rough milling 1: 0.1 nA rough milling 2: 50 pA lamella polishing: 10 or 30 pA at a stage tilt of 15° (milling angle of 8°).

### Cryo-EM data collection

Lamellae were imaged using a Titan Krios 300 keV cryo-TEM (Thermo Fisher) equipped with a K3 direct detector (Gatan) and an energy filter (Gatan), slit width 20 eV at a magnification of 81000 x, corresponding to a sample pixel size of 1.06 Å, and a 100 µm objective aperture. A defocus of –0.5 µm was targeted using an adjacent sacrificial area and the autofocus function in SerialEM (*Mastronarde, 2005*). Movies were collected at an exposure rate of 1 e⁻/Å²/frame to a total dose of 30 e⁻/Å².

## Image processing

Images were processed using *cis*TEM (*Grant et al., 2018*) as described previously (*Lucas et al., 2021*), and using sample tilt determination implemented in a modified version of CTFFIND4 (*Lucas et al., 2021*; *Rohou and Grigorieff, 2015*) to estimate sample defocus and to account for the ~8° tilt of the lamella introduced during FIB-milling. Images of 3D densities and 2DTM results were prepared in ChimeraX (*Pettersen et al., 2021*).

## 2DTM

The molecular models noted in the text were aligned to one another to have the same origin using their 28S rRNA using the MatchMaker function in UCSF Chimera (*Meng et al., 2006*; *Pettersen et al., 2004*) and 2DTM templates were generated by simulating 3D densities (*Himes and Grigorieff, 2021*). 2DTM was performed using the program *match_template* in the *cis*TEM GUI (*Lucas et al., 2021*) using the default parameters. Significant targets were defined as those exceeding a threshold of 7.85, which allows a single false positive per image, assuming no correlation between the rotational or defocus searches. Prior comparisons with 3DTM results suggest that this threshold is conservative and that the true false positive rate is likely lower (*Lucas et al., 2021*). The coordinates were refined using the program *refine_template* (*Lucas et al., 2021*) in rotational steps of 0.1° and a defocus range of 200 Å with a 10 Å step.

## 3D reconstruction using mature 60S 2DTM coordinates

We used the program *prepare_stack_matchtemplate* (*Lucas et al., 2021*) to generate a particle stack using the locations and orientations of the significant mature 60S-detected targets after refinement as described above. We then used *cis*TEM to generate a 3D reconstruction from 3991 mature 60S targets detected in 28 images of the nuclear periphery, only including targets with a 2DTM SNR of >8. The reconstruction had a nominal resolution of 3.5 Å using an Fourier Shell Correlation (FSC) threshold of 0.143 and a mask radius of 175 Å (*Figure 1—figure supplement 3A*; *Rosenthal and Henderson, 2003*) that is expected to overestimate the resolution due to overfitting (*Grigorieff, 2000*; *Lucas et al., 2021*). To best capture the density in the 40S, we low-pass filtered the reconstruction to 10 Å, representing an FSC of 0.9. Local resolution calculations were performed using *local_resolution* in Phenix (*Liebschner et al., 2019*), default parameters and extending to 3.5 Å.

## Calculating 2DTM SNR values and ratios of SNR values

Targets identified in two or more searches with aligned templates were identified using the program *align_coordinates* (*Lucas et al., 2021*). The 2DTM SNRs of targets identified in two or more searches were compared by taking the $\log_2$ of the SNR ratio. The $\log_2$ was used in place of the direct ratio because the shape of the distribution is independent of the order of comparison, except for a mirror around 0, while the distribution of the direct ratios shows more complicated behavior. Histograms of both the $\log_2$ values and direct ratios of the cytoplasmic 60S population have approximately Gaussian distributions with fits characterized by the coefficient of determination $R^2=0.993$ and $R^2=0.991$ respectively. To calculate the change in the 2DTM SNR with modified templates, the program *refine_template* (*Lucas et al., 2021*) was used to calculate 2DTM SNRs for additional templates using the locations and orientations from a previous exhaustive search with an initial template, without performing a rotational search by specifying the rotational step as 360°. To obtain consistent ratios of 2DTM SNRs, the 2DTM SNR values for both the initial template and the additional templates were calculated.

## Calculating histograms from different subcellular regions

Subregions of the image in *Figure 1B*, corresponding to the nucleus, cytoplasm and vacuole were identified as indicated in *Figure 1—figure supplement 2*. Care was taken to avoid the edges of the image and regions of the image corresponding to membranes. The corresponding regions of the normalized MIP resulting from a 2DTM search using the mature 60S template were extracted using the clip resize function in IMOD (*Kremer et al., 1996*), defining a box of 1000x1000 pixels. Image histograms were calculated from the flattened array of pixel values extracted using the python suite mrcfile (*Burnley et al., 2017*) using only the 2DTM SNR values below the calculated threshold.

## Calculating relative probabilities

Histograms were generated (bin 0.05) of the calculated $\log_2$ 2DTM SNR ratios and Gaussians were fitted using GaussianMixture in sklearn (*Pedregosa et al., 2011*). Based on the shape of the histogram,

we model the $log_2$ 2DTM SNR ratios as a mixture of $K$-component multivariate Gaussian distributions, when $K$ templates are used in the search. We fit Gaussians to the $log_2$ SNR ratios of any two selected templates. Each target $i$ is then associated with $K-1$ such SNR ratios $x_i$. For example, for $K = 4$, we can define the following:

$$X_i = \begin{bmatrix} log_2(SNR_{i,k=1}/SNR_{i,k=2}) \\ log_2(SNR_{i,k=1}/SNR_{i,k=3}) \\ log_2(SNR_{i,k=1}/SNR_{i,k=4}) \end{bmatrix} \tag{2}$$

For particles belonging to the same population (class), the $log_2$ SNR ratio can be described by the multivariate Gaussian probability density function (PDF):

$$P(X_i|\Theta_k, Z_i = k) \sim \mathbb{N}(M_k, \Sigma_k) = \frac{1}{(2\pi)^{\frac{d}{2}} |\Sigma_k|^{\frac{1}{2}}} \exp\left(\frac{(X_i - M_k)^T \Sigma_k^{-1}(X_i - M_k)}{2}\right) \tag{3}$$

$$P(z_i = k) = \pi_k \tag{4}$$

where $X_i$ is a vector of $K-1$ $log_2$ SNR ratios, $z_i$ indicates the identity of the target ($k = 1, 2, \ldots, K$), and $\Theta_k = \{M_k, \Sigma_k, \pi_k\}$ is the set of parameters of the Gaussian PDF $N$ and the prior probability that a detected target belonging to class $k$. The total joint likelihood for $N$ detected targets is then

$$L(\Theta; X) = P(X|\Theta) = \Pi_{i=1}^N P(X_i|\Theta) = \Pi_{i=1}^N \Sigma_{j=1}^K \pi_j N(M_j, \Sigma_j) \tag{5}$$

with $\Theta = \{\Theta_1, \Theta_2 \ldots \Theta_K\}$ and $X = \{X_1, X_2 \ldots X_N\}$.

We use an expectation-maximization (EM) algorithm to iteratively calculate the maximum likelihood estimates of the model parameters where the E-step calculates the posterior probability via Bayes rule,

$$P(z_i = k|X_i, \Theta) = \frac{\pi_k \mathbb{N}(M_k, \Sigma_k)}{\Sigma_{j=1}^K \pi_j \mathbb{N}(M_j, \Sigma_j)} \tag{6}$$

and the M-step updates the model parameters for each class,

$$\pi_k = \frac{\Sigma_{i=1}^N P(z_i = k|X_i, \Theta)}{N} \tag{7}$$

$$M_k = \frac{\Sigma_{i=1}^N X_i \cdot P(z_i = k|X_i, \Theta)}{\Sigma_{i=1}^N P(z_i = k|X_i, \Theta)} \tag{8}$$

$$\Sigma_k = \frac{\Sigma_{i=1}^N P(z_i = k|X_i, \Theta)(X_i - M_k)(X_i - M_k)^T}{\Sigma_{i=1}^N P(z_i = k|X_i, \Theta)} \tag{9}$$

Prior probabilities ($\pi$) can be set by subjective assessment based on the experiment, or set to $1/K$ where all classes have equal probability. For example, to determine the relative probability that an LN 60S-detected nuclear target belongs to the LN 60S or EN 60S class, we assume that their relative frequencies are the same and therefore the prior probability of the two intermediates in the nucleus is equal: $P(LN\,60S) = (EN\,60S) = 0.5$.

## Simulations

We used the *cis*TEM program *simulate* (**Himes and Grigorieff, 2021**) to simulate images of 200 randomly oriented LN or mature 60S using a pixel size of 1.06 Å, an ice thickness of 100 nm amd an exposure rate of 1 e⁻/Å² to a total dose of 30 e⁻/Å², matching the experimental data. 2DTM was performed as described above.

## Acknowledgements

We are grateful to Michael Rosbash (Brandeis) for providing MNY8 cells, Xiaowei Zhao for help trouble-shooting FIB-milling, the cryo-EM facility at Janelia Research Campus where this data was collected and Peter Rickgauer, Tim Grant and Ben Himes for helpful comments and suggestions. This project has been made possible in part by grant number 2021–234617 from the Chan Zuckerberg Initiative DAF, an advised fund of Silicon Valley Community Foundation awarded to NG and BL.

## Additional information

### Competing interests
Nikolaus Grigorieff: Reviewing editor, *eLife*. The other authors declare that no competing interests exist.

### Funding

| Funder | Grant reference number | Author |
|---|---|---|
| Chan Zuckerberg Initiative | 2021-234617 | Bronwyn A Lucas<br>Nikolaus Grigorieff |
| Howard Hughes Medical Institute | | Nikolaus Grigorieff |

The funders had no role in study design, data collection and interpretation, or the decision to submit the work for publication.

### Author contributions
Bronwyn A Lucas, Conceptualization, Data curation, Formal analysis, Funding acquisition, Validation, Investigation, Visualization, Methodology, Writing - original draft, Project administration, Writing – review and editing; Kexin Zhang, Software, Formal analysis, Validation, Visualization, Methodology, Writing – review and editing; Sarah Loerch, Methodology, Writing – review and editing; Nikolaus Grigorieff, Conceptualization, Software, Formal analysis, Supervision, Funding acquisition, Methodology, Writing - original draft, Project administration, Writing – review and editing

### Author ORCIDs
Bronwyn A Lucas  http://orcid.org/0000-0001-9162-0421
Sarah Loerch  http://orcid.org/0000-0002-1731-516X
Nikolaus Grigorieff  http://orcid.org/0000-0002-1506-909X

### Decision letter and Author response
Decision letter https://doi.org/10.7554/eLife.79272.sa1
Author response https://doi.org/10.7554/eLife.79272.sa2

---

## Additional files

### Supplementary files
• MDAR checklist

### Data availability
Micrographs, templates and scaled maximum intensity projections (MIPs) in this study have been deposited to EMPIAR and are accessible with the following public access code: EMPIAR-10998. Micrographs, templates and normalized maximum intensity projections (MIPs) in this study are accessible with the following public access code: EMPIAR-10998. A compiled snapshot of the code used for 2D template matching and tilted CTF estimation is available to download from https://cistem.org/development.

The following dataset was generated:

| Author(s) | Year | Dataset title | Dataset URL | Database and Identifier |
|---|---|---|---|---|
| Lucas B, Zhang K, Loerch S, Grigorieff N | 2022 | In situ single particle classification reveals distinct 60S maturation intermediates in cells | https://www.ebi.ac.uk/empiar/EMPIAR-10998/ | EBI, EMPIAR-10998 |

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
