## [Editor Report]

This paper explores the use of 2D high-resolution template-matching (2DTM) to locate and discriminate highly similar macromolecules within cryo-EM images of focused ion beam-milled cells. It demonstrates that differences in the 2DTM signal-to-noise ratios for located targets against multiple search templates can effectively segregate a mixed population of similar structures, as well as present a formal analysis strategy for probabilistic assignment of species within the mixed population. Because the identification of distinct structural states of macromolecular complexes inside the cell is a fundamental problem in 3D visual proteomics, this paper will be of broad interest to both structural and cell biologists.

---

## [Decision Letter]

**Decision letter after peer review:**

Thank you for submitting your article "In situ single particle classification reveals distinct 60S maturation intermediates in cells" for consideration by *eLife*. Your article has been reviewed by 3 peer reviewers, including Sjors HW Scheres as Reviewing Editor and Reviewer #1, and the evaluation has been overseen by Anna Akhmanova as the Senior Editor. The following individual involved in review of your submission has agreed to reveal their identity: Thomas G Laughlin (Reviewer #3).

Given the close relationship between this paper and *eLife* paper https://elifesciences.org/articles/68946, I propose to change the current submission to a Research Advance, directly linking the two. Could you please do so when you re-submit your revised paper?

Essential revisions:

(1) One major concern that needs to be addressed more openly in a revised version of the paper is that of potential model bias of incorrect references. The observation on page 18 that the reconstruction from the picked particles was estimated by FSC to be 3.5A, yet the resulting map had to be filtered at 10A to limit the noise, is a strong indication that model bias does play a role in the identification of particles. This bias must affect the measured SNR scores, and thus the probabilistic metrics presented. It also suggests that part of the identified picks may in fact not be true 60S ribosomes, but false positives. This would then affect the conclusions drawn. If the authors disagree, they should set out clear arguments for their case. Also, they should discuss how potential overfitting or model bias would affect their new metrics for particle classification in the discussion. Currently, the only reference to the dangers of overfitting is on page 16, merely referring to their previous paper.

Further questions to consider in this light are:

(a) It is hard to tell from figure 1H how good the density for the 40S is, but from the figure it looks as though it is lower quality than the 60S. Unless there is substantial variation in the relative positions of 40S and 60S, then this is possibly explained by some amount of reference bias? Can the author comment on this? Related: is the FSC in Fig1Suppl1D unmasked (including 40S)?

(b) In figure 2, the assignment of ribosomes in the cytoplasm is quite convincing, however the template matching results are more ambiguous for the nuclear ribosomes. The authors propose this is because in the nucleus there are multiple states (later revisited in Figure 5). But can they exclude part of the ambiguity is due to the nucleus being a more 'dense' environment, generating more noisy data?

Can the authors exclude false positives have been picked particularly in the nucleus? One good control would be to generate reconstructions of ribosomes picked from the nucleus using either template and assessing the 40S density.

(c) Ln 167: What is the expected false-positive rate per image based on the chosen SNR threshold? One per image?

(2) The authors have been wise in selecting the ribosome as a test case. Probably, because of its size and RNA content, for many instances of this complex the SNR is sufficiently high for detection/classification. However, if less careful authors would choose a smaller target, what would happen? What would be the pitfalls and how could they be avoided? This needs a more open discussion in the paper, in order to obtain better insights into the general usefulness of the methodology.

(3) There is no information on the thickness of the lamellae. This would be useful information, and if a range of thicknesses was used, whether there is any correlation between thickness and the ability of 2DTM to discriminate between classes of ribosomes? Does the defocus/Z-refinement of the templates indicate a minimal apparent thickness?

(4) Also, assuming a thickness between 100 and 200 nm, this would mean between a quarter and one eight of ribosomes would be 'cut' by the milling. Have the authors considered whether and how this might affect their analysis?

(5) Regarding the ML approach to establish probability of matched particles to belong to a certain class. In the case reported here priors were assigned based on location. For Figure 5, priors for classification of the two nuclear populations are assigned assuming equal distribution. It is not clear how important these decisions are on the outcome of the classification. In the vast majority of cases there will be no spatial distinction between particles with similar structures, nor will we have any idea about their relative frequencies. How applicable is this methodology to cases other than nuclear versus cytoplasmic ribosomes? I wonder whether a plot like the one in 3B could be derived from in silico data, where the relative abundance is exactly known, and then used to derive probabilities for 'real' data?

(6) Figure 4D could be interpreted in the following way: when the template only contains RNA, the SNR decreases more for nuclear and less for cytoplasmic particles, meaning that the protein part of the LN template contributed less to the 'discovery' of cytoplasmic ribosomes than nuclear. Which makes sense. However, I am not sure I understand the meaning of the experiment on the right side of the panel: when there is only protein (of the LN template), the SNR generally decreased indicating the RNA played an important role in detecting particles – but it decreases more for cytoplasmic ribosomes. This seems to me to suggest that the LN RNA contributed more to detecting the cytoplasmic ribosomes than the nuclear ones, which I struggle to understand. In other words: I agree with lines 245-247 (quote: "the LN60S proteins contribute more to the SNR of the nuclear targets than the cytoplasmic targets and are therefore more effective at differentiating the nuclear from the cytoplasmic 60S populations"), but if that is true then one would be equally able to state that: "the LN60S RNA contributes more to the SNR of the cytoplasmic targets than the nuclear targets and are therefore more effective at differentiating the nuclear from the cytoplasmic 60S populations, but in the wrong way around". Can the authors please explain this?

(7) Lines 428-435. I find this concept interesting, however it would be a challenge to find the most 'comprehensive' template without trying all the available ones. For example, in the case discussed here, if the idea was applied using the mature 60S template the results would have been possibly quite similar, however using the LN 60S template would have significantly skewed the analysis.

(8) Lines 459-464. The idea of refining atomic models against 2D images of cellular samples would be quite revolutionary, however to implement it there would be more required than just "addressing potential of overfitting" and "detecting and quantifying errors in the refined model". I suggest to remove the paragraph.

(9) Ln 110: How many lamellae were the 28 images acquired from?

(10) Figure 2 —figure supplement 1: What is being indicated by the different height lines under panel D? Could this be indicated in the legend.

*Reviewer #1 (Recommendations for the authors):*

Page 8: The sentences about a threshold defining so many percent of targets were not entirely clear to me. Perhaps they could be reworded?

Page 10: the first sentence seems rather far-reaching for the data presented, and might be wrongly interpreted by less careful users of the technique.

Page 14: "A major challenge … limit radiation damage": a verb is missing?

Figure 1E (and others) just look very black. Perhaps a colour figure would be easier to print/look at? It would also be insightful to have a colour scale bar next to it.

Page 18 mentions a modified version of CTFFIND4. Is this openly available?

*Reviewer #2 (Recommendations for the authors):*

1. It is hard to tell from figure 1H how good the density for the 40S is, but from the figure it looks as though it is lower quality than the 60S.

Unless there is substantial variation in the relative positions of 40S and 60S, then this is possibly explained by some amount of reference bias? Can the author comment on this?

Related: is the FSC in Fig1Suppl1D unmasked (including 40S)?

2. There is no information on the thickness of the lamellae. This would be useful information, and if a range of thicknesses was used, whether there is any correlation between thickness and the ability of 2DTM to discriminate between classes of ribosomes?

3. Also, assuming a thickness between 100 and 200 nm, this would mean between a quarter and one eight of ribosomes would be 'cut' by the milling.

Have the authors considered whether and how this might affect their analysis?

4. In figure 2, the assignment of ribosomes in the cytoplasm is quite convincing, however the template matching results are more ambiguous for the nuclear ribosomes. The authors propose this is because in the nucleus there are multiple states (later revisited in Figure 5). But can they exclude part of the ambiguity is due to the nucleus being a more 'dense' environment, generating more noisy data?

Can the authors exclude false positives have been picked particularly in the nucleus?

One good control would be to generate reconstructions of ribosomes picked from the nucleus using either template and assessing the 40S density.

5. Regarding the ML approach to establish probability of matched particles to belong to a certain class. In the case reported here priors were assigned based on location. For Figure 5, priors for classification of the two nuclear populations are assigned assuming equal distribution. It is not clear how important these decisions are on the outcome of the classification. In the vast majority of cases there will be no spatial distinction between particles with similar structures, nor will we have any idea about their relative frequencies. How applicable is this methodology to cases other than nuclear versus cytoplasmic ribosomes? I wonder whether a plot like the one in 3B could be derived from in silico data, where the relative abundance is exactly known, and then used to derive probabilities for 'real' data?

6. Figure 4D could be interpreted in the following way: when the template only contains RNA, the SNR decreases more for nuclear and less for cytoplasmic particles, meaning that the protein part of the LN template contributed less to the 'discovery' of cytoplasmic ribosomes than nuclear. Which makes sense. However, I am not sure I understand the meaning of the experiment on the right side of the panel: when there is only protein (of the LN template), the SNR generally decreased indicating the RNA played an important role in detecting particles – but it decreases more for cytoplasmic ribosomes. This seems to me to suggest that the LN RNA contributed more to detecting the cytoplasmic ribosomes than the nuclear ones, which I struggle to understand.

In other words: I agree with lines 245-247 (quote: "the LN60S proteins contribute more to the SNR of the nuclear targets than the cytoplasmic targets and are therefore more effective at differentiating the nuclear from the cytoplasmic 60S populations"), but if that is true then one would be equally able to state that: "the LN60S RNA contributes more to the SNR of the cytoplasmic targets than the nuclear targets and are therefore more effective at differentiating the nuclear from the cytoplasmic 60S populations, but in the wrong way around". Can the authors please explain this?

7. Lines 428-435. I find this concept interesting, however it would be a challenge to find the most 'comprehensive' template without trying all the available ones. For example, in the case discussed here, if the idea was applied using the mature 60S template the results would have been possibly quite similar, however using the LN 60S template would have significantly skewed the analysis.

8. Lines 459-464. The idea of refining atomic models against 2D images of cellular samples would be quite revolutionary, however to implement it there would be more required than just "addressing potential of overfitting" and "detecting and quantifying errors in the refined model". I suggest to remove the paragraph.

*Reviewer #3 (Recommendations for the authors):*

Comments and questions:

Ln 110: How many lamellae were the 28 images acquired from?

Can the authors comment on the thicknesses of lamellae images? Does the defocus/Z-refinement of the templates indicate a minimal apparent thickness?

Ln 167: What is the expected false-positive rate per image based on the chosen SNR threshold? One per image?

Figure 2 —figure supplement 1: What is being indicated by the different height lines under panel D? Could this be indicated in the legend.

I commend the authors for deposition of all requisite material to reproduce the study to EMPIAR. In addition, could the authors elaborate more on the Cryo-EM data collection for reference for others intending to use this approach for their samples of interest (e.g., target defoci, filter slit-width, number of frames, etc.).

---

## [Author Response]

Essential revisions:(1) One major concern that needs to be addressed more openly in a revised version of the paper is that of potential model bias of incorrect references. The observation on page 18 that the reconstruction from the picked particles was estimated by FSC to be 3.5A, yet the resulting map had to be filtered at 10A to limit the noise, is a strong indication that model bias does play a role in the identification of particles. This bias must affect the measured SNR scores, and thus the probabilistic metrics presented. It also suggests that part of the identified picks may in fact not be true 60S ribosomes, but false positives. This would then affect the conclusions drawn. If the authors disagree, they should set out clear arguments for their case. Also, they should discuss how potential overfitting or model bias would affect their new metrics for particle classification in the discussion. Currently, the only reference to the dangers of overfitting is on page 16, merely referring to their previous paper.

We thank the reviewers for encouraging us to elaborate on this point. We agree that template bias and the potential for false positives are both important considerations. We have updated the manuscript to include further discussion of how these factors could affect our analysis as well as additional control experiments.

As the reviewers point out, our reconstruction suffers from template bias that results from partial alignment of the background with the template. This may indeed affect the observed SNR values of detected targets. However, this will not change the rate of false positive detections, which depends on the distribution of correlation coefficients expected for pure noise, and the chosen threshold, which was set to one false positive per search in our experiments. For the reconstruction, we only allowed targets above an SNR threshold of 8, which would allow five false positives of the 3991 targets (0.1%). We therefore do not expect that false positives affect the reconstruction. Consistently, we have shown previously that we are able to recover features in reconstructions using 2DTM coordinates that do not derive from the template (Rickgauer et al., 2017, Lucas et al., 2021) and that we do not reproduce template features in the reconstruction that are not in the data (Lucas et al., 2021). This indicates that reconstructions using 2DTM coordinates are not dominated by template bias.

We have included a more thorough discussion of template overfitting to noise and how it affects the results under the subheading “Relative similarity to alternate templates reveals population identity”. We further elaborate on these points in the discussion on page 18, lines 535-547 and page 19, lines 569-584.

Further questions to consider in this light are:(a) It is hard to tell from figure 1H how good the density for the 40S is, but from the figure it looks as though it is lower quality than the 60S. Unless there is substantial variation in the relative positions of 40S and 60S, then this is possibly explained by some amount of reference bias? Can the author comment on this? Related: is the FSC in Fig1Suppl1D unmasked (including 40S)?

Indeed, there is substantial heterogeneity in the position of the 40S relative to the 60S. The text has been updated to reflect this point (page 5, lines 150-156). Additional discussion has been included in the manuscript: page 18, lines 534-546. The Materials and methods have been updated to include the mask radius on page 22, line 679.

(b) In figure 2, the assignment of ribosomes in the cytoplasm is quite convincing, however the template matching results are more ambiguous for the nuclear ribosomes. The authors propose this is because in the nucleus there are multiple states (later revisited in Figure 5). But can they exclude part of the ambiguity is due to the nucleus being a more 'dense' environment, generating more noisy data?Can the authors exclude false positives have been picked particularly in the nucleus? One good control would be to generate reconstructions of ribosomes picked from the nucleus using either template and assessing the 40S density.

Differences in background in different parts of the image may affect observed 2DTM SNR values, and thus particle classification (see above). However, due to the whitening and normalization procedures, the false-positive rates remain approximately constant across an image. We now include Figure 1—figure supplement 2 to show that the histogram of normalized MIP values is comparable in the nucleus and the vacuole relative to the cytoplasm, and if anything, shifted slightly to the left. Therefore, we would expect that, relative to the cytoplasm, the probability of a false positive in the nucleus, is similar or slightly lower. This is also visible in Figure 2D-G, which shows little difference in the background in the nucleus relative to the cytoplasm and between the two templates in the small area shown, and in Figure 3B, which shows that the standard deviations of the log_2_(2DTM SNR ratios) in the nucleus and the cytoplasm are similar, indicating similar noise levels. Consistently, the vacuole is much darker in many images than the nucleus. However, of the 4601 independent 60S-detected targets in the searches with the three different templates, only one was found in the vacuole (0.02%). We have included a discussion of this point in the manuscript on pages 4, line 125 – page 5, line 131 and in the discussion on page 19, lines 569-584.

We thank the reviewers for the suggested control experiment to detect 40S bound to 60S targets detected in the nucleus. However, nuclear maturation of 60S and 40S occurs separately, and therefore we would not expect pre-60S to be associated with 40S. As a control, we showed in the original manuscript that inhibiting 60S export depletes targets detected with the EN 60S template.

(c) Ln 167: What is the expected false-positive rate per image based on the chosen SNR threshold? One per image?

We have updated the Materials and methods to include this information.

(2) The authors have been wise in selecting the ribosome as a test case. Probably, because of its size and RNA content, for many instances of this complex the SNR is sufficiently high for detection/classification. However, if less careful authors would choose a smaller target, what would happen? What would be the pitfalls and how could they be avoided? This needs a more open discussion in the paper, in order to obtain better insights into the general usefulness of the methodology.

Thank you for raising this issue. We have included a discussion of this point in the manuscript page 19, line 578-584.

(3) There is no information on the thickness of the lamellae. This would be useful information, and if a range of thicknesses was used, whether there is any correlation between thickness and the ability of 2DTM to discriminate between classes of ribosomes? Does the defocus/Z-refinement of the templates indicate a minimal apparent thickness?

We have updated the manuscript to include a table describing the image-wise thickness estimates and defocus.

(4) Also, assuming a thickness between 100 and 200 nm, this would mean between a quarter and one eight of ribosomes would be 'cut' by the milling. Have the authors considered whether and how this might affect their analysis?

Any factors that decrease the agreement between the template and the target would reduce the 2DTM SNR and increase the standard deviation of the log_2_(2DTM SNR ratios), making classification more challenging (e.g.: Figure 3—figure supplement 2). We do not expect that any cut or partial ribosomes are detected. While the sample may contain partial ribosomes on the surface of the lamella, interaction with the 30 kV Ga+ beam would likely cause significant damage and degradation of their structure, making them unlikely to be detected. Damage further inside the sample would also be expected to lower the 2DTM SNRs by varying degrees, again leading to increased standard deviations and lower 2DTM SNRs.

(5) Regarding the ML approach to establish probability of matched particles to belong to a certain class. In the case reported here priors were assigned based on location. For Figure 5, priors for classification of the two nuclear populations are assigned assuming equal distribution. It is not clear how important these decisions are on the outcome of the classification. In the vast majority of cases there will be no spatial distinction between particles with similar structures, nor will we have any idea about their relative frequencies. How applicable is this methodology to cases other than nuclear versus cytoplasmic ribosomes? I wonder whether a plot like the one in 3B could be derived from in silico data, where the relative abundance is exactly known, and then used to derive probabilities for 'real' data?

The ability to distinguish the nuclear from the cytoplasmic populations provides a convenient control for testing our approach in situ. We agree that in most cases the subcellular localization will not distinguish classes. This is the justification for the experiment in Figure 5, wherein we attempt to distinguish a mixed population of nuclear intermediates without prior information about their spatial distribution. In this case, we use a 50:50 prior for the two nuclear intermediates, effectively assuming no prior information. We were able to identify changes to the relative occupancy of nuclear 60S populations after treatment with a nuclear export inhibitor. We conclude that this strategy can be used to classify mixed populations with different relative occupancy.

The purpose of this experiment was to show that we could distinguish populations in cells that differ with respect to their relative similarity to different templates. As the reviewers point out, an *in silico* experiment could be used to show that log_2_(2DTM SNR ratios) can be used to separate related states. Indeed, we show in the new Figure 4 that provided sufficient targets are selected, the mean log_2_(mature 60S/ LN60S 2DTM SNR) is stable across a wide range of 2DTM SNRs. We have described how this can be used to validate the identity of a population by comparison with population means from simulations. However, extrapolating from distributions fit to simulated data requires accurately simulating the in situ noise. We show in the new Figure 4 that differences in background alter the standard deviation of the fitted populations and therefore affect classification, making direct extrapolation of probabilities unreliable. We suggest, therefore, that to accurately capture the noise, populations will need to be identified by fitting distributions to the data in each experiment. In the case wherein the populations are mixed, we included biological controls to show that we could detect expected changes to populations in the cell.

(6) Figure 4D could be interpreted in the following way: when the template only contains RNA, the SNR decreases more for nuclear and less for cytoplasmic particles, meaning that the protein part of the LN template contributed less to the 'discovery' of cytoplasmic ribosomes than nuclear. Which makes sense. However, I am not sure I understand the meaning of the experiment on the right side of the panel: when there is only protein (of the LN template), the SNR generally decreased indicating the RNA played an important role in detecting particles – but it decreases more for cytoplasmic ribosomes. This seems to me to suggest that the LN RNA contributed more to detecting the cytoplasmic ribosomes than the nuclear ones, which I struggle to understand. In other words: I agree with lines 245-247 (quote: "the LN60S proteins contribute more to the SNR of the nuclear targets than the cytoplasmic targets and are therefore more effective at differentiating the nuclear from the cytoplasmic 60S populations"), but if that is true then one would be equally able to state that: "the LN60S RNA contributes more to the SNR of the cytoplasmic targets than the nuclear targets and are therefore more effective at differentiating the nuclear from the cytoplasmic 60S populations, but in the wrong way around". Can the authors please explain this?

Thank you for highlighting this point. Indeed, this result seems counterintuitive at first. We have updated our description of this experiment in the text.

(7) Lines 428-435. I find this concept interesting, however it would be a challenge to find the most 'comprehensive' template without trying all the available ones. For example, in the case discussed here, if the idea was applied using the mature 60S template the results would have been possibly quite similar, however using the LN 60S template would have significantly skewed the analysis.

Indeed, using the LN 60S as the template for the “discovery” search would likely miss mature 60S targets, thereby biasing the number detected targets towards the LN 60S population. If the SNR values for the templates are too different, a full search may have to be performed for all of them to avoid such a bias.

(8) Lines 459-464. The idea of refining atomic models against 2D images of cellular samples would be quite revolutionary, however to implement it there would be more required than just "addressing potential of overfitting" and "detecting and quantifying errors in the refined model". I suggest to remove the paragraph.

We have removed this paragraph.

(9) Ln 110: How many lamellae were the 28 images acquired from?

We have included this information in the text on page 4, line 113. We have additionally included a table of image characteristics, including estimated thickness and defocus calculated using CTFFIND.

(10) Figure 2 —figure supplement 1: What is being indicated by the different height lines under panel D? Could this be indicated in the legend.

The lines indicate images from the same lamella. We have included this information in the figure legend.